# Isotopic evidence concerning the habitat of *Nautilus macromphalus* in New Caledonia

**Amane Tajika** [1,2]*, **Neil H. Landman**[1], **J. Kirk Cochran**[1,3], **Claire Goiran**[4], **Aubert Le Bouteiller**[5]

1 Division of Paleontology (Invertebrates), American Museum of Natural History, New York, NY, United States of America, 2 University Museum, University of Tokyo, Tokyo, Japan, 3 School of Marine and Atmospheric Sciences, Stony Brook University, Stony Brook, NY, United States of America, 4 Université de la Nouvelle Calédonie, Nouméa, New Caledonia, 5 N˚34, rue Thomy Célières, Nouméa, New Caledonia

* atajika@amnh.org

**Data Availability Statement:** All relevant data are within the manuscript and its Supporting Information.

**Funding:** AT was supported by a Grant-in-Aid for JSPS Research Fellow, Grant-in-Aid for Young

## Abstract

Modern nautilids (*Nautilus* and *Allonautilus*) have often been studied by paleontologists to better understand the anatomy and ecology of fossil relatives. Because direct observations of these animals are difficult, the analysis of light stable isotopes (C, O) preserved in their shells has been employed to reveal their habitat and life history. We aim to (1) reconstruct the habitat depth of *Nautilus macromphalus* and (2) decipher the fraction of metabolic carbon in its shell by analyzing oxygen and carbon isotopes ($\delta^{18}O$, $\delta^{13}C$) in the septa of two specimens in combination with analyses of water samples from the area. Additionally, we investigate whether morphological changes during ontogeny are reflected in the isotopic values of the shells. Results reveal that the patterns of change of $\delta^{18}O$ and $\delta^{13}C$ in the septa of *N. macromphalus* pre- and post-hatching are consistent with previous studies. Values of $\delta^{18}O_{water}$ range from 0.7 to 1.4‰ (VSMOW), with a maximum value coincident with a salinity maximum at ~150 m. We use the temperature and $\delta^{18}O_{water}$ profiles to calculate equilibrium values of $\delta^{18}O_{aragonite}$ with depth. Comparing these values with the measured $\delta^{18}O$ of the septa shows that the habitat depth of *N. macromphalus* is ~140 m pre-hatching and ~370 m post-hatching. Using $\delta^{13}C$ of shell carbonate and published data on metabolic carbon, the fraction of metabolic carbon is reconstructed as ~21% and 14% pre- and post-hatching, respectively. The reconstructed depth pre-hatching is slightly shallower than in *N. pompilius* from the Philippines and Fiji, but the post-hatching depth is similar. However, it is important to emphasize that these estimates represent average over time and space because nautilus is a mobile animal. Lastly, the changes in morphological parameters and the changes in $\delta^{13}C$ and $\delta^{18}O$ during ontogeny do not coincide except at hatching and at the onset of maturity.

## Introduction

*Nautilus* is an iconic marine mollusk. It is the only externally shelled cephalopod alive today. It is a member of a vast group of shelled cephalopods, including ammonites and nautiloids, that once inhabited the planet. It comprises two genera *Nautilus* and *Allonautilus*. Based on

Scientists (grant nrs. 20J00376 and 21K14028). NHL and JKC were supported by the Norman D. Newell Fund (AMNH). The funders had no role in study design, data collection and analysis, decision to publish, or preparation of the manuscript.

**Competing interests:** The authors have declared that no competing interests exist.

morphological and molecular data, the two genera include at least eight species, which are restricted to isolated archipelagos across the Indo-Pacific [1].

Because nautilus live in deep fore-reef environments, direct observations of their habitat and life history have been difficult. Nevertheless, expeditionary research in the last 50 years has yielded spectacular results. Efforts to capture and release animals, track their movements using radio transmitters attached to the shells, and remote cameras have produced a complex picture of their behavior within their habitat [2–4]. These animals are nektobenthic, meaning that they generally live just above the sea bottom, and make vertical migrations along the slope [2–4].

Another approach to throw light on the habitat and life history of these animals is the analysis of stable isotopes of carbon and oxygen preserved in the shell. The isotope ratios $^{18}O/^{16}O$ and $^{13}C/^{12}C$, expressed as delta values ($\delta^{18}O$, $\delta^{13}C$) relative to a standard (PeeDee Belemnite, PDB), can provide information on the temperature of the water [5–7], and on the isotope composition of carbon in the dissolved inorganic carbon reservoir ($\delta^{13}C$) in which the shell formed [8, 9].

Analyses of oxygen and carbon isotopes are very useful in understanding living nautilus, but they are invaluable in reconstructing the life history and habitat of fossil nautilids and ammonites. Because direct observations of these animals are impossible, the use of oxygen and carbon isotopes offer important insights, provided that the shells are well enough preserved and have not suffered diagenetic alteration [10]. Combined with clues from facies distribution, faunal association, and fossil preservation, analysis of the oxygen and carbon isotope composition of the shells can shed light on their habitat and rate of the growth of these extinct organisms [11–15].

The use of these methods requires knowledge of the chemistry ($\delta^{18}O$ of water, $\delta^{13}C$ of dissolved inorganic carbon, DIC) and temperature of the water from the site the animals inhabit. Although such information is difficult to assemble for the geologic past, it is surprisingly absent in most studies of modern nautilus (only example: *Nautilus pompilius* from the Philippines and Fiji [16]). This is due to the fact that trapping for nautilus and collecting water samples at the same time is a difficult proposition and that the analysis of $\delta^{18}O$ of water and $\delta^{13}C$ of DIC requires subsequent gas-source mass spectrometry. As a result, one usually relies on published values from worldwide ocean databases, which may not be detailed enough for the specific sites where nautilus live.

While the $\delta^{18}O$ of the shell allows us to reconstruct the water temperature and thus habitat depth of nautilus, the $\delta^{13}C$ of the shell has been considered difficult to interpret as a proxy for paleoenvironment. This is because the $\delta^{13}C$ of the shell is both a function of carbon incorporated via the metabolism of the animal as well as a function of the dissolved inorganic carbon (DIC) [17, 18]. Elucidating the fraction of metabolic carbon in the $\delta^{13}C$ of the shell is of importance because it can permit the reconstruction of the DIC of the ancient oceans as a new proxy.

As documented previously, the isotopic composition of the shell changes during ontogeny [16, 19–24]. Do these changes coincide with changes in morphology? Such knowledge is of relevance when reconstructing the ecology of extinct cephalopods where the shells are sometimes not well enough preserved for isotope analysis. Although highly resolved morphological examination of nautilid conchs was difficult in the past, it is now possible owing to the advancement of tomographic methods. Using such methods, classical morphological parameters such as the whorl expansion rate and the whorl width index as well as the siphuncle position index at various ontogenetic points can be measured [25].

In this study, we aim to answer the following questions: (1) What is the habitat depth of *Nautilus macromphalus* through ontogeny? (2) What is the fraction of metabolic $\delta^{13}C$ in the

nautilid shell? (3) Are changes in $\delta^{18}O$ and $\delta^{13}C$ of nautilid shells reflected in changes in morphology?

## Background

Based on studies of nautilus raised in aquaria, nautilus secretes its shell in oxygen isotope equilibrium with its environment [23]. Thus, the $\delta^{18}O$ of the nautilus shell can be used as a reliable indicator of the temperature of the environment in which it lives. There are no indications of a "vital effect". This applies to the septa secreted in the egg capsule as well as those secreted post-embryonically. In contrast, as noted above, the $\delta^{13}C$ of the shell is a function of fractionation between it and the $\delta^{13}C$ of the dissolved inorganic carbon (DIC), as well as the incorporation of metabolic carbon from food sources.

Septa are convenient targets of isotopic analysis. However, they do not form instantaneously but over a finite interval of time. This time interval increases over ontogeny. Landman and Cochran [26], in a review of nautilus growth rates, estimated that the length of septal secretion in early ontogeny, following hatching, may approach weeks, but at the onset of maturity, it may approach months. Thus, the isotope compositions of individual septa reflect time averages.

During embryonic development, the animal secretes its shell in an egg capsule attached to the sea floor. In aquaria, embryonic development takes as long as one year [27]. Following hatching, nautilus is a mobile animal. Based on previous studies, it is clear that it migrates vertically along the slope of the steep fore-reef [28]. These migrations may occur every day or every few days [29]. In addition, nautilus migrates laterally, sometimes for long distances. Saunders and Spinosa [4] recorded movements of as much as 150 km in 332 days around Palau, Western Caroline Islands. Thus, the isotopic compositions of septa reflect averages over space as well as time.

Furthermore, nautilus is long-lived, perhaps as long as 15 or even 20 years [26, 30], and the environment in which it lives may change over this time period. Therefore, the temperature and salinity recorded at a single marine station may not encompass the changes in the environment during the lifetime of the animal. On the other hand, prior studies [28] suggest that nautilus mostly lives below 50 m depth, except for rare reports [31]. Thus, seasonal variation in the water column, which is limited mainly to the mixed layer, may not be a determining factor in the interpretation of isotopic values.

Finally, analyses of the outer shell wall of nautilus have revealed a range of isotope values, probably reflecting depth migration [29]. Similarly, analyses through the thickness of an individual septum reveal patterns of isotopic variation, also probably reflecting vertical migration [16]. Based on these results, it appears that nautilus secretes its shell more-or-less continuously throughout its life at whatever depth it inhabits. Thus, not only is $\delta^{18}O$ in nautilus shell material a reliable indicator of the environment in which it forms its shell (i.e., no vital effect), but shell secretion is continuous, unlike that, for example, in intertidal bivalves.

In summary, the oxygen isotope composition of septa secreted in the egg capsule during embryonic development represents a single depth and temperature over a period of as much as one year. In contrast, that of the post-hatching septa represent averages. These averages do not imply a single preferred depth, but instead integrate over time and space. In other words, they represent the depths (and temperatures) that nautilus inhabit, on average.

## Materials and methods

### Sample collection

Two specimens (one male and one female) of *Nautilus macromphalus* Sowerby (1849) were captured live in 2002 using traps baited with fish at a depth of 400 m off Nouméa, New

Caledonia (Fig 1). The male specimen (AMNH 132423) is regarded as mature because it exhibits septal approximation (i.e., closer spacing of septa) and a black band at the aperture. These features are diagnostic of maturity [32]. The female specimen (AMNH 132380) is most likely submature considering that the septa are approximated, but the black band is absent. The specimens were sectioned along the median plane, and pieces were removed from each septum through ontogeny (with septum 1 as the first formed septum) for carbon and oxygen isotope analysis. The samples represented the entire thickness of a septum at any point, but the complete septum was not sampled or homogenized.

Water samples were collected in June, 2003, from two stations (Station 25 and 49; Fig 1A) near the site at which the nautilus specimens were captured. Samples (~1L) were taken every 50 m (to 400 m) and stored sealed in glass bottles in the dark for oxygen isotope measurement (see below). In addition, temperature and salinity of the water were measured every half-meter using a CTD.

## Oxygen and carbon isotope analysis

Oxygen and carbon isotopes of the septal samples were measured at the University of California Santa Cruz Stable Isotope Laboratory. Pieces of each septum were broken off and sent to the laboratory. Prior to analysis, samples were ground to a size of 125–250 μm and 40–60 μg of sample were selected for analysis. Standards and samples were conditioned in a 60°C vacuum-roasted oven overnight to remove any residual water from the glass or samples. Samples were then analyzed for $\delta^{18}O$ and $\delta^{13}C$ via acid digestion using an individual vial acid drop Thermo-Scientific Kiel IV carbonate device interfaced to a ThermoScientific MAT-253 dual-inlet isotope ratio mass spectrometer (IRMS). Samples were reacted at 75°C in orthophosphoric acid (specific gravity = 1.92 g/cm$^3$) to generate carbon dioxide and water. Non-condensable gases were pumped away, and the $CO_2$ analyte was then cryogenically separated from water, finishing with the introduction of pure $CO_2$ into the IRMS via the dual inlet.

Raw data were also corrected using a two-point calibration against samples of calibrated in-house granular Carrara Marble standard reference material ($\delta^{13}C$ = 2.05 ± 0.1‰ and $\delta^{18}O$ = -1.91 ± 0.1‰ VPDB) and granular NBS-18 limestone international standard reference material. The in-house Carrara Marble was extensively calibrated against NIST Standard Reference Material (NBS-19, NBS-18, and LSVEC) and further calibrated in intercomparison studies with international laboratories. Raw data were corrected for offset from the international standard PDB (PeeDee Belemnite) for $\delta^{18}O$ and $\delta^{13}C$ and corrected for instrument-specific source ionization effects. Aliquots of an external working standard (powdered Atlantis II calcium carbonate; $\delta^{18}O$ = 3.42‰) were run "as-a-sample" to monitor quality control and long-term performance. Typical precision was 0.05‰ (1σ) for both $\delta^{18}O$ and $\delta^{13}C$. All values were reported relative to VPDB.

A few samples of septa were sent to the University of Michigan Stable Isotope Laboratory for cross-comparison. These were of two types: intact pieces of septa and pieces that were ground before submission to the laboratory. Carbonate samples weighing a minimum of 10 micrograms of pure carbonate were reacted with anhydrous phosphoric acid in a Thermo Kiel IV preparation device coupled directly to the inlet of a Thermo MAT 253 triple collector isotope ratio mass spectrometer. $^{17}O$-corrected data were subsequently corrected for acid fractionation and source mixing by calibration to a best-fit regression line defined by two NBS standards, NBS 18 and NBS 19. Data were reported in ‰ notation relative to VPDB. Precision and accuracy of data were monitored through the analysis of a variety of powdered carbonate standards. Measured precision was maintained at better than 0.1 ‰ for both carbon and oxygen isotope compositions. These precisions comprise uncertainties of ~4% of the measured

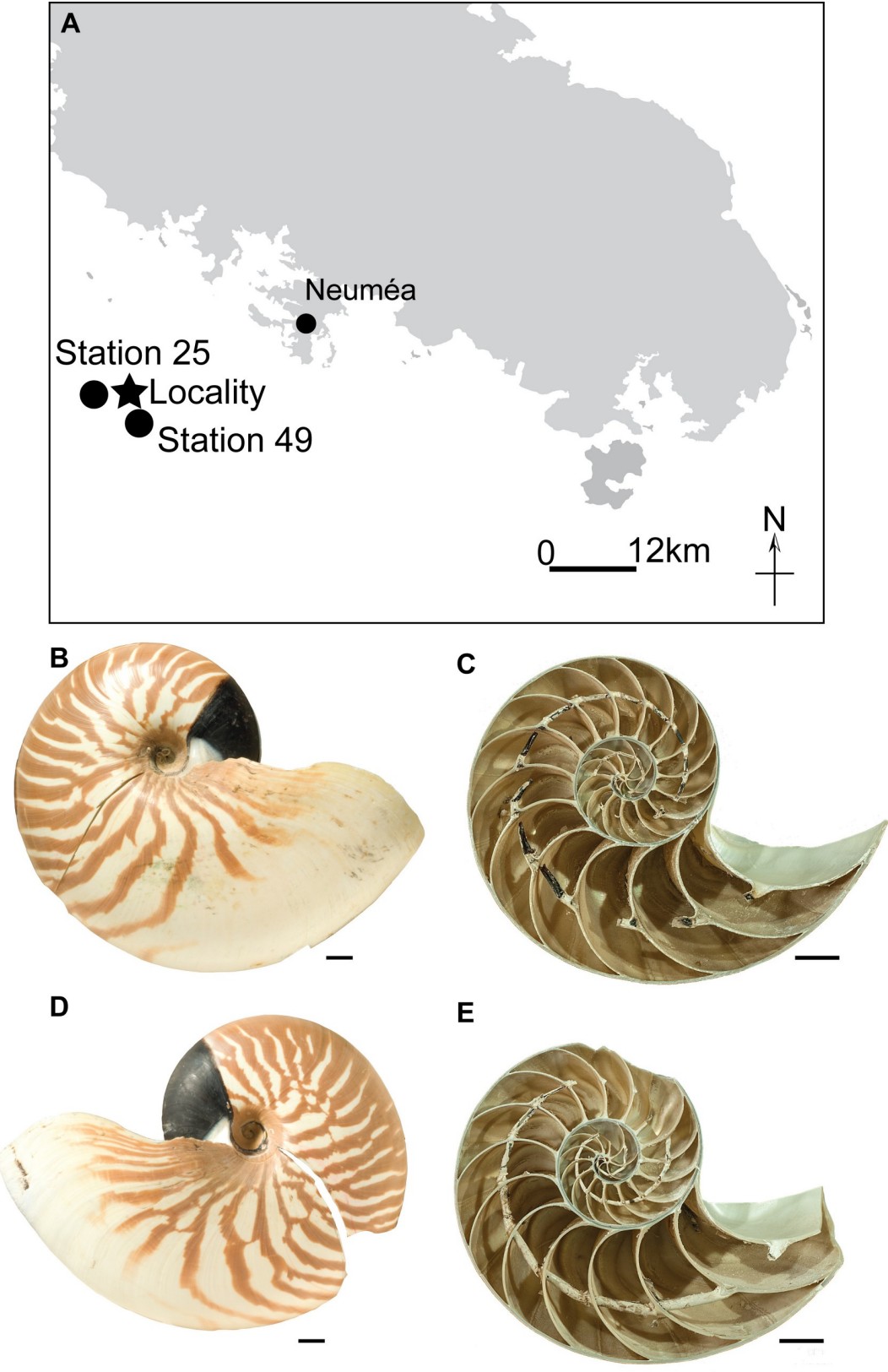

**Fig 1. Study area and specimens.** Map of New Caledonia. Star indicates the locality at which the specimens were caught. Circles indicate two stations at which water samples were taken (A); Outer shell of *Nautilus macromphalus* (AMNH 132380;

female) (B); Cross-section of *N. macromphalus* (AMNH 132380; female) (C); Outer shell of *N. macromphalus* (AMNH 132423; male) (D); Cross-section of *N. macromphalus* (AMNH 132423; male) (E). Scale bars = 10 mm. Samples were taken through ontogeny (the earliest formed septum as septum 1). Map was reproduced from OpenStreetMap (© OpenStreetMap contributors).

septal $\delta^{18}O$ values and 9–18% for the $\delta^{13}C$ values (S1 Table). The discrete water samples obtained at Station 25 and 49 (Fig 1A) were analyzed for $\delta^{18}O$ of the water at the Keck Paleoenvironmental and Environmental Stable Isotope Laboratory. The water samples were not poisoned with $HgCl_2$ after collection to prevent bacterial activity, which could alter the DIC and $\delta^{13}C_{DIC}$. Thus, we do not report $\delta^{13}C_{DIC}$ values and, alternatively, we use literature values of $\delta^{13}C_{DIC}$ appropriate to the study site, as described in detail below. For oxygen isotope measurements, aliquots of samples were equilibrated with $CO_2$; $\delta^{18}O_{water}$ was determined using a ThermoFinnigan MAT 253 mass spectrometer. Values were reported relative to VSMOW with an internal precision of ±0.02‰.

## Morphological analysis

To examine the morphology of the two conchs of *Nautilus macromphalus* through ontogeny, we CT-scanned the specimens with a voxel size of ~0.66 mm at the Microscopy and Imaging Facility of the American Museum of Natural History. The CT-scans obtained were used to measure the following morphological parameters (Fig 2): conch diameter (dm), whorl width (ww), whorl height (wh), distance between the ventral edge of the siphuncle and the ventral edge of the conch (vd) and septal spacing (rotational angle). We took these measurements every 30˚ (rotational angle; starting at aperture) for dm, ww, and wh, while vd was measured at each septum. These parameters were used to calculate the whorl expansion rate $[(dm_1/dm_2)^2;$ WER], whorl width index (ww/dm; WWI), whorl height, and siphuncle position index (vd/wh; SPI). For details of these morphological parameters, see Tajika and Klug [25], and Tajika et al. [33]. Direct comparisons between WER, WWI, and the isotope values are difficult because they require estimation of the position of the aperture at the time of septal formation. According to Collins and Ward [32], the body chamber angle increases by approximately 15˚ after the formation of the last septum. Therefore, we estimated the position of the aperture by subtracting 15˚ from the observed body chamber length (compare with Ohno et al. [19]). In contrast, direct comparisons were possible between septal spacing, SPI, and the isotope values. We calculated Spearman's rank correlation coefficients to determine if the morphological and isotopic values were correlated.

## Ethics statement

The specimens of *Nautilus macromphalus* (Mollusca: Cephalopoda) used in this study were captured live in a baited trap by Dr. Royal H. Mapes in 2002 before the species was given local protected status around New Caledonia in 2008, and before they were protected under the Convention on International Trade in Endangered Species of Wild Fauna and Flora (CITES) in 2016. The individuals were euthanized and retained for morphological study. The specimens are reposited at the American Museum of Natural History.

## Results

### Hydrography

Temperature (T) and salinity (S) profiles in the upper 400 m of the water column are shown in Fig 3A and 3B. The temperature profiles show surface water with ~23˚C, extending over a ~30

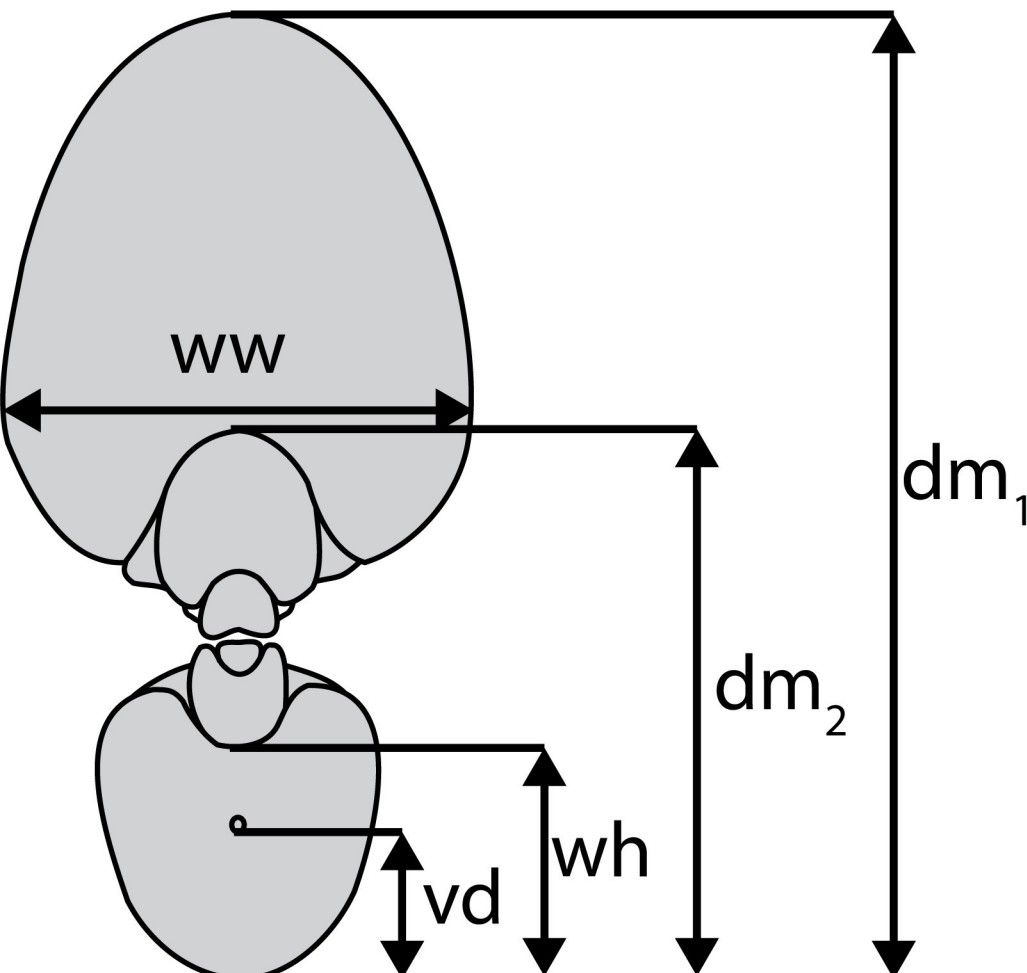

**Fig 2. Cross-section of *Nautilus macromphalus* with measured morphological conch parameters.** ww = whorl width, dm = conch diameter, wh = whorl height, ww = whorl width, and vd = distance between the ventral edge of the siphuncle and the ventral edge of the conch.

m mixed layer. T values decrease to ~13°C at 400 m. Salinity is ~35.4 (as psu) in surface water, increases to 35.6 at ~150–200 m and then decreases to 35.1 at 400 m. A T-S plot of the data (Fig 3C) shows mixing among three water masses: 1) surface water (S~35.5; T~23°C), 2) sub-surface salinity maximum water (S ~35.65; T~19°C, and 3) deep water (S ~35.1; T~13°C). These measurements represent a single snapshot of the water column whereas the lifetime of *Nautilus macromphalus* extends over at least 10 years [30]. However, annual seasonal variation in temperature and salinity is largely confined to the upper 50 m of the water column [34]. Previous studies of *N. macromphalus* indicate that it usually lives in deeper water [28], suggesting that seasonal variation may be of little importance in interpreting the isotopic signals in the shell.

## Isotope hydrography

The water column isotope data are presented in S1 Table. The $\delta^{18}O_{water}$ profiles are similar at the two stations (Fig 4), showing values increasing from ~0.85‰ at 50 m to values of 0.91–1.38‰ at the depth of the salinity maximum and then decreasing to 0.6–0.8‰ at ~400 m

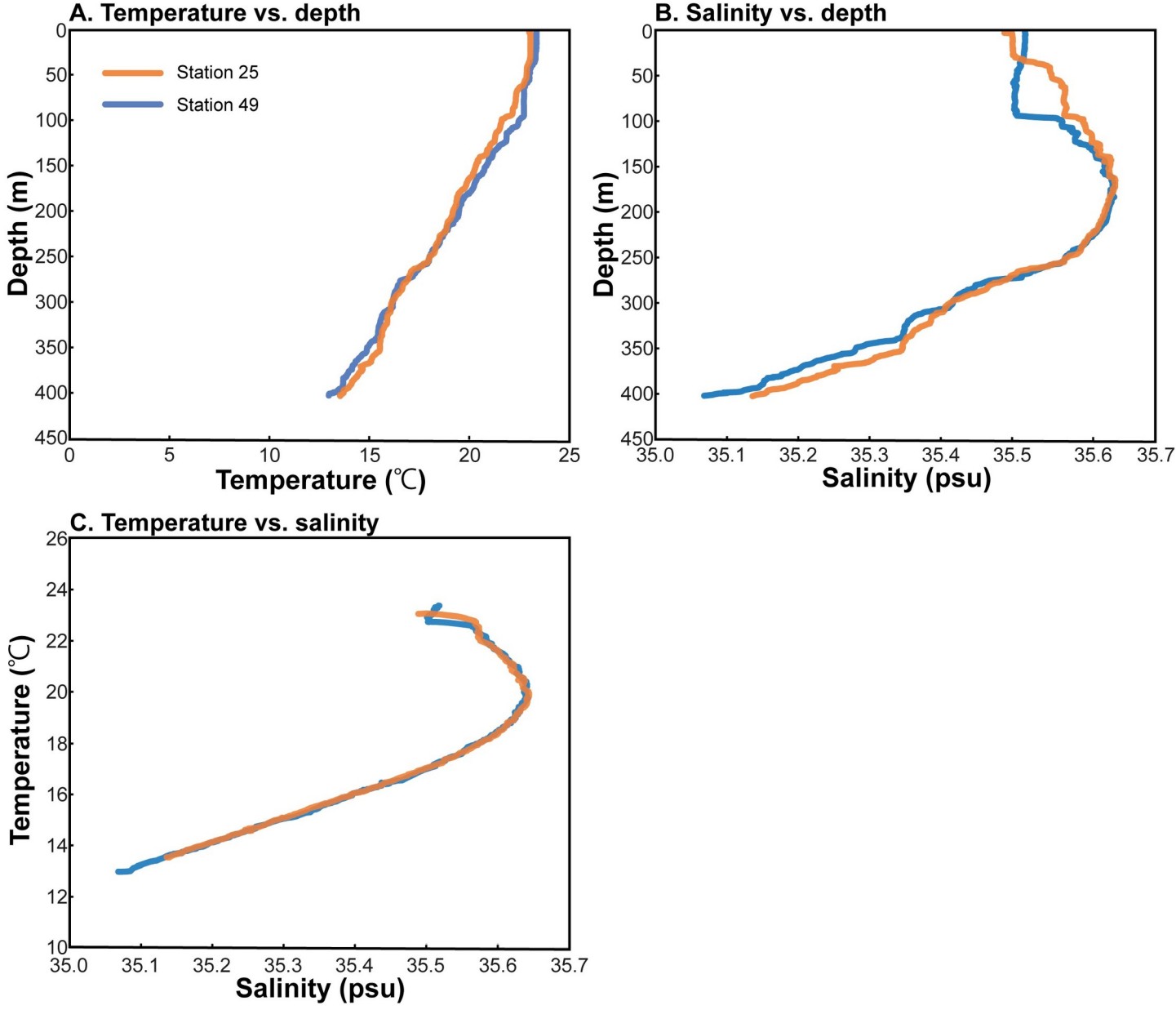

**Fig 3. Hydrography.** Temperature versus depth (A); Salinity versus depth (B); TS diagram (C).

(S1 Table). The integrated water column (50–400 m) average $\delta^{18}$O is 0.93‰. As mentioned above, the water samples were not poisoned with $HgCl_2$ and consequently, values of $\delta^{13}C_{DIC}$ were not reliable. Thus, we use literature values of $\delta^{13}C_{DIC}$ [35] to interpret the septal carbonate $\delta^{13}$C data, as described in detail below.

## C and O isotopes in the septa

The trends of C and O isotope data of the septa are similar in the two specimens (Fig 5; S1 Table). Both show lower $\delta^{18}$O values ranging from 0.32 to 0.73‰ in septa 1–7, increasing to 1.13 to 2.35‰ after septum 7. Both specimens show lower $\delta^{18}$O in the two most recently

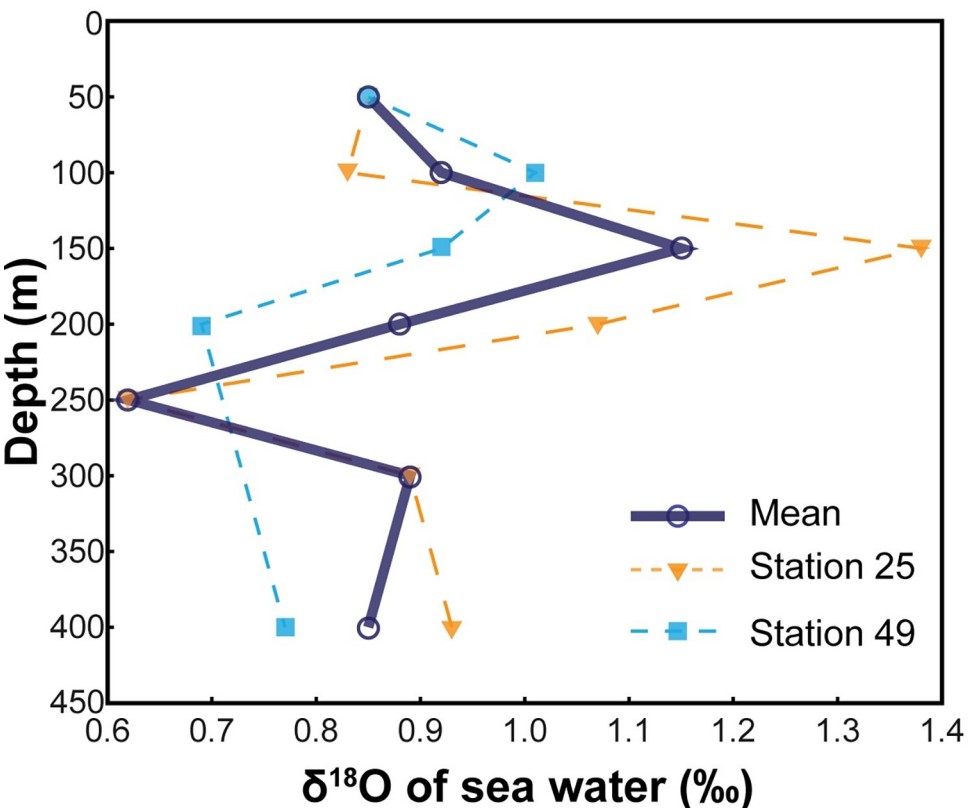

**Fig 4. Isotope hydrography.** Oxygen isotope composition ($\delta^{18}O_{water}$; ‰ SMOW) versus depth.

deposited septa. Values of $\delta^{13}C$ progressively increase in the first 7–8 septa of each specimen, and then decrease for the next two septa and increase again through ontogeny, varying from -0.01 to 1.49‰. Results of the laboratory intercomparison are presented in Fig 5 and S1 Table. Given that different pieces of a particular septum were analyzed by the two laboratories, the

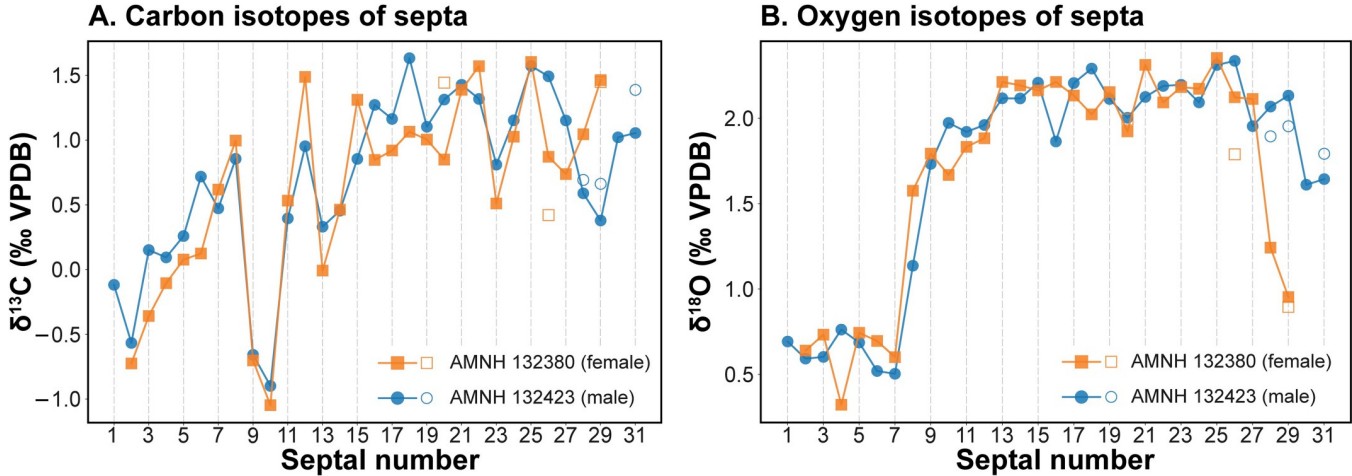

**Fig 5. Septal C and O isotopes.** $\delta^{13}C$ (‰ VPDB) (A); $\delta^{18}O$ (‰ VPDB) (B). Filled markers represent the data produced at the University of California, Santa Cruz. Unfilled markers represent the data produced at the University of Michigan.

general agreement is within 10–20%. One sample (AMNH 13423, septum 28) showed a value from Santa Cruz that was ~50% greater than that from Michigan, but when the ground sample was re-analyzed at Michigan, it agreed within 0.10‰ for both $\delta^{18}O$ and $\delta^{13}C$ (S1 Table). We consider this comparison further below in the context of reconstructing the habitat depth of *Nautilus macromphalus* from the oxygen isotope data.

### Morphology

The morphological parameters in both specimens of *Nautilus macromphalus* are shown in Fig 6. Remarkable changes occur between pre- and post-hatching in all morphological parameters. Septal spacing increases and then decreases rapidly at hatching and it decreases moderately thereafter until septum 14 in the female (AMNH 132480) and septum 17 in the male (AMNH 132423). In the female specimen, the septal spacing fluctuates with a more or less constant mean value of 24˚ (Fig 6A). In contrast, the male specimen shows an increase after septum 17 and the angle remains the same (24˚) until it starts decreasing at septum 25 (Fig 6B). Septal spacing decreases to a lower value at the last septum (male: 22.6˚; female: 20.6˚).

The siphuncle position index increases rapidly until septum 14 with a plateau between about septa 5 and 8 (Fig 6C and 6D). Then, it shows a gradually decreasing trend until the end of ontogeny. A somewhat rapid decrease in the siphuncle position index is visible before the attainment of maturity in both the female and male.

The whorl expansion rate (WER) shows a sharp decrease until the point of hatching (~30 mm), which is followed by an increase (Fig 6E and 6F). During the juvenile stage, the female specimen exhibits a higher WER than the male (about 3.2), which gradually decreases until a diameter of ~ 90 mm. Thereafter, WER increases until the end of ontogeny. The male shows a WER of ~ 3.0 after hatching until a diameter of ~ 80 mm after which it starts increasing. At about 110 mm, WER becomes stable until the end of ontogeny. The whorl width index (WWI) rapidly decreases until the end of the embryonic stage, and then the decrease becomes moderate. The moderate decrease in WWI stops at about 75 mm in both specimens. While the female exhibits a stable WWI after a conch diameter of 80 mm, the male shows an increasing trend. WWI abruptly decreases at the end of ontogeny in the female and male.

## Discussion

### Oxygen isotopes in the septa: Habitat depth

Previous studies have used the classic equation relating the $\delta^{18}O$ of molluscan aragonite to the temperature and $\delta^{18}O$ of the water in which it was secreted. For example, the equation of Grossman and Ku [6], as modified by Hudson and Anderson [36], such that the $\delta_{water}$ values are corrected for the difference between the PDB and SMOW scales, is:

$$T(^{\circ}C) = 19.7 - 4.34(\delta_{aragonite} - \delta_{water}) \tag{1}$$

As noted in the Background section, Landman et al. [23] analyzed the shells of two specimens of *Nautilus belauensis* that grew in aquaria with measured water temperatures. The eggs were laid in the aquarium and developed for nearly one year. They hatched and lived for several months after hatching. The calculated water temperatures from the oxygen isotope analysis of the shells showed good agreement with the measured temperatures in the aquarium tanks, demonstrating that they were secreted in isotopic equilibrium with seawater, both before and after hatching. Indeed, the results reported by Landman et al. [23] provided a foundation for the paleotemperature interpretation of the oxygen isotope record of fossil nautilids and ammonites.

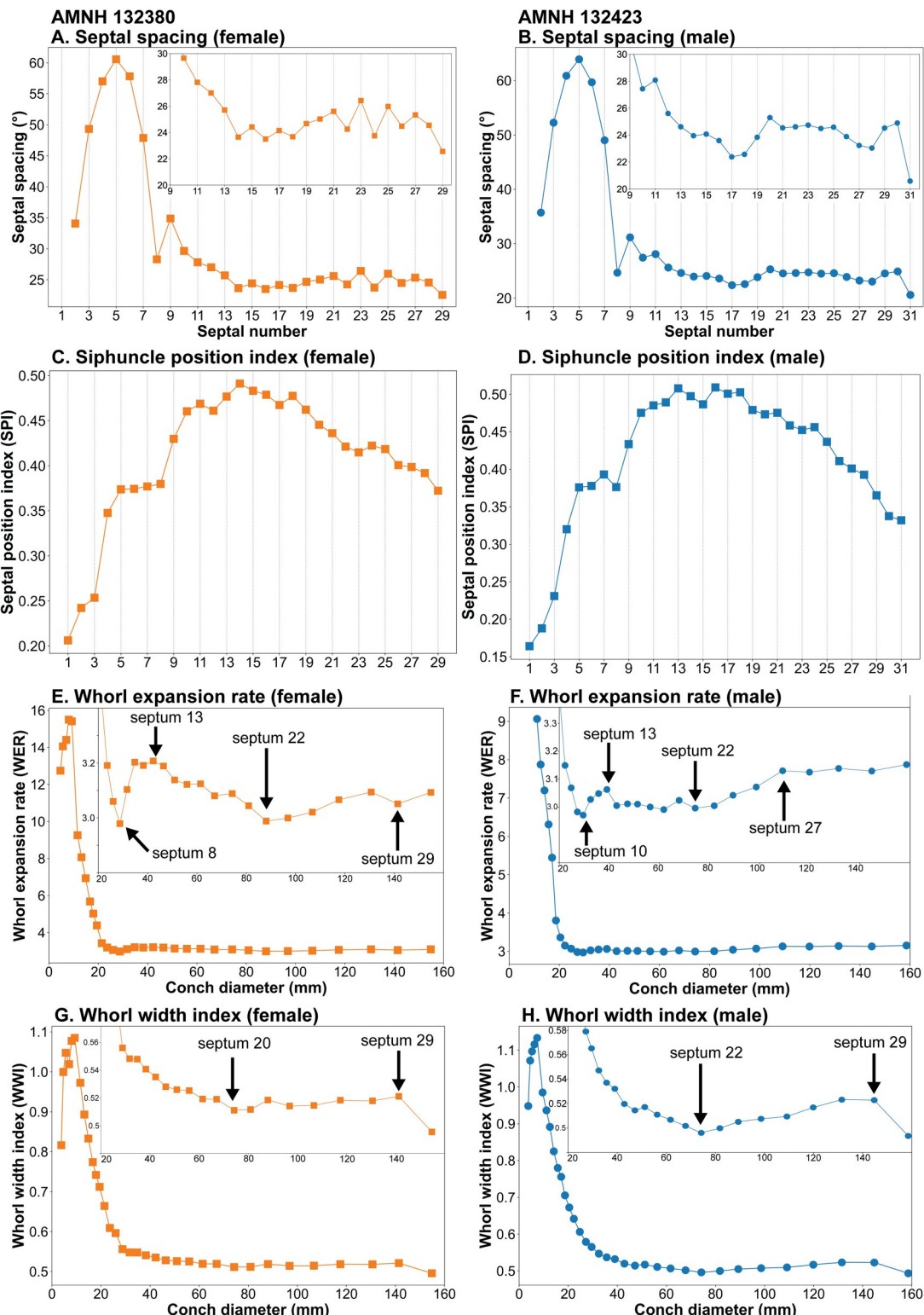

**Fig 6. Morphological parameters in *Nautilus macromphalus*.** Septal spacing for AMNH 132380 (female) (A); septal spacing for AMNH 132423 (male) (B); siphuncle position index (= vd/wh) for AMNH 132380 (female) (C); siphuncle position index (= vd/wh) for AMNH 132423 (male) (D); whorl expansion rate [= $(dm_1/dm_2)^2$] for AMNH 132380 (female) (E); whorl expansion rate [= $(dm_1/dm_2)^2$] for AMNH 132423 (male) (F); whorl width index (= ww/dm) for AMNH 132380 (female) (G); whorl width index (= ww/dm) for AMNH 132423 (male) (H).

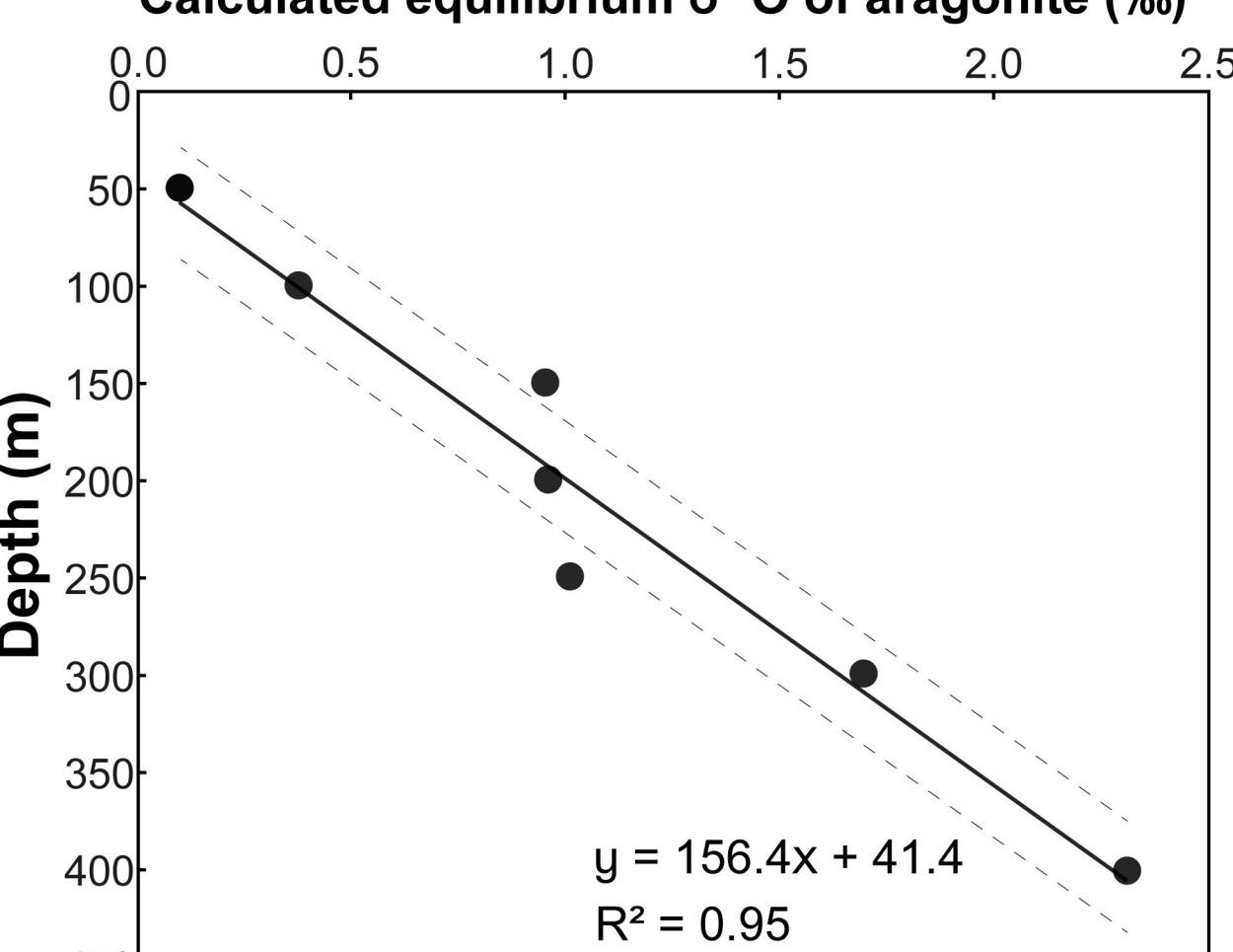

**Fig 7. Calculated δ¹⁸O$_{aragonite}$ vs. depth.** Dashed lines indicate the 95% confidence interval.

In the present study, we have measured the depth profiles of both temperature and $\delta^{18}O$ of the water in which the nautilus were caught. We can thus rewrite Eq 1 to calculate the $\delta^{18}O$ of precipitated biogenic aragonite ($\delta^{18}O_{ar}$) as a function of the two measured variables, and, in effect, as a function of depth in the water column (S1 Table):

$$\delta^{18}O_{ar} = -\left(\frac{(T(^{\circ}C) - 19.7)}{4.34}\right) + \delta^{18}O_w \qquad (2)$$

Fig 7 shows how the calculated $\delta^{18}O_{ar}$ varies with depth in the water column. We use the linear relationship between depth and $\delta^{18}O$ of precipitated aragonite to determine the depths of precipitation of each septum. The results reveal shallower depths for septa deposited pre-hatching. Indeed, average $\delta^{18}O$ values for the pre- and post-hatching septa (S1 Table; Fig 5) yield average depths of 140 ± 8 m and 370 ± 25 m, respectively (Fig 8), with corresponding water temperatures of ~21°C and ~15°C, respectively. The shaded areas in Fig 8 indicate the estimated errors in depth calculated from the 95% confidence interval of the regression line in Fig 7. Although the estimated error produces some variation in depth (~±30 m), the two

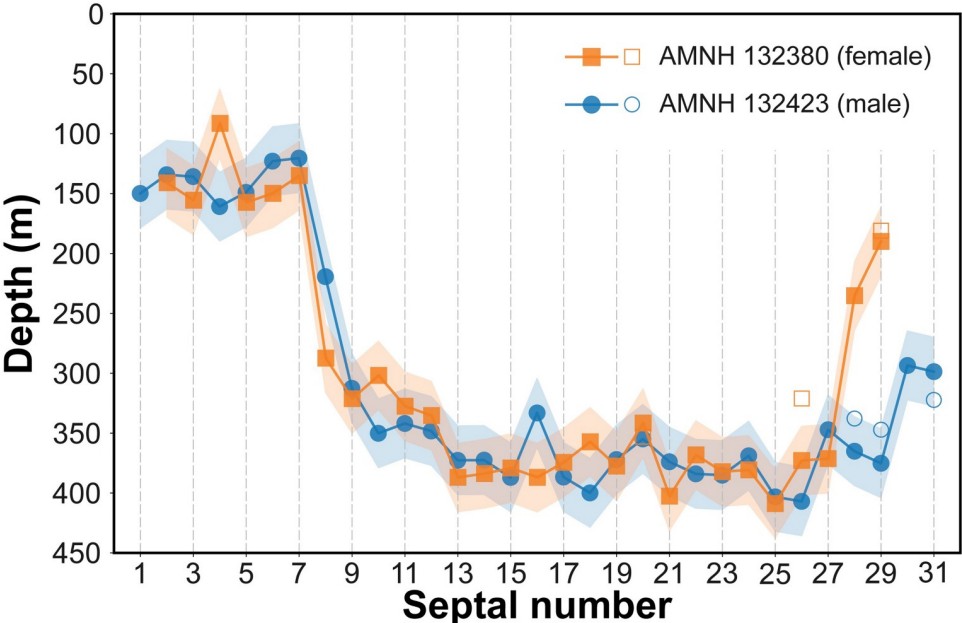

**Fig 8. Reconstructed habitat depths of *Nautilus macromphalus* from New Caledonia.** Filled markers represent the data produced at the University of California, Santa Cruz. Unfilled markers represent the data produced at the University of Michigan. Shaded areas represent the estimated error range calculated with the 95% confidence interval in Fig 7.

specimens show the same pattern during ontogeny. The shallower depth for septa deposited pre-hatching within the egg capsule is consistent with prior studies [16, 21–23, 37] that show warmer temperatures and shallower depths for egg incubation in modern nautilids.

We indicate in Fig 8 the difference in the calculated depth that results from our intercomparison of $\delta^{18}O$ values measured at different laboratories (S1 Table). In general, the difference is small, although in one case (AMNH 132423, septum 28), the averages of the $\delta^{18}O$ measurements of the septal piece and ground sample run at the Michigan Stable Isotope Facility are 1.39‰ and 2.02, yielding depths of 258 m and 357 m, respectively. The measurement from the Santa Cruz Isotope Facility of this septum (2.07‰) yields a depth of 365 m. The multiple measurements from the different labs likely reflect variations related to our sampling method (small pieces of each septum). The variation is consistent with the observation of Oba et al. [16], Ohno et al. [19], and Zakharov et al. [37] who documented variations of 0.3–0.6‰, 0.9‰, and 0.4‰, respectively in individual septa from *Nautilus pompilius* collected in the Philippines. As noted in the Background section, septa do not form instantaneously but over a time interval that increases through ontogeny [21, 26, 38]. The differences in $\delta^{18}O$ in single septa are consistent with movements of nautilus through the water column over time.

For the septa deposited post-hatching, the depth calculated from $\delta^{18}O$ is consistent with the trapping depth of the specimens in New Caledonia (400 m). However, given that the time of secretion of each septum increases through ontogeny [21, 26, 38], the calculated depths and temperatures based on analyses of the entire septal thicknesses represent temporal averages. The exact time for septal formation is not clear, but observations in aquaria and inferences based on radiometric data in *N. macromphalus* suggest 2–3 weeks for juvenile and > 10 weeks for sub-adult/adult individuals [30, 39].

Temporal variation is also evident in the outer shell of *N. macromphalus* from New Caledonia, as shown in the detailed measurements by Secondary Ion Mass Spectrometry (SIMS) by

Linzmeier et al. [29] in a specimen from the same collection as the specimens in the present study. The results show that the animal migrated over depths representing a range of $\delta^{18}O_{ar}$ 2.5‰. Using ultrasonic telemetry techniques, Dunstan et al. [2] have shown that *N. pompilius* in Australia can range over depths bounded by the upper limit of temperance tolerance (~25˚C) and the implosion depth of the shell (~800 m). The 2.5‰ range in $\delta^{18}O$ measured by Linzmeier et al. [29] brackets the range of our calculated $\delta^{18}O_{ar}$ (translating to 50 to 400 m depth), but could include deeper depths as the temperature decreases and the calculated $\delta^{18}O_{ar}$ increases with depth to ~3.5‰ at 800 m.

It is notable that the $\delta^{18}O$ of the last two septa in both specimens from New Caledonia reflect shallower depths. The male is mature and the female is submature. This is expressed in the approximation of the last two septa in both specimens. It is likely that the migration to shallower and warmer water of the female is linked to selection of hatching sites for incubation. It is also possible that, with respect to both sexes, mating preferentially occurs in shallower water.

We compare our results with the data of Davies et al. [40] on a specimen of *Nautilus macromphalus* from the same collection as our specimens. Davies et al. [40] measured both conventional $\delta^{18}O$ and carbonate clumped isotopes ($\Delta47$). They measured $\delta^{18}O$ in selected post-hatching septa, with values ranging from 0.73 to 1.47‰ VPDB. The lower values were observed in the most recently deposited septa, similar to our observations. Davies et al. [40] used an estimated $\delta^{18}O_w$ of 0.5‰ to calculate growth temperatures of ~12–14˚C. Recalculating their results using our measured higher $\delta^{18}O_w$ yields estimated depths of precipitation ranging from ~160–265 m with temperatures of ~17.48–20.60˚C. Interestingly, these values are more comparable, although still lower than the range of clumped isotope temperatures of ~17–31˚C calculated by Davies et al. [40] on the same specimen.

As noted above, the age of maturity of *Nautilus macromphalus* is at least 10–12 years, and the period of septal formation increases over ontogeny from weeks in early ontogeny to months at the onset of maturation [30]. Given that our water samples were collected at one time, the septa we analyzed did not form in the exact seawater we sampled. Previous studies documented temporal and seasonal variation of seawater temperature and chemistry at/near New Caledonia [34, 41]. Van Den Broeck et al. [34] documented a seasonal change in temperature and salinity of ~3˚C and ~0.3 (psu), respectively, in the upper 50 m. At deeper depths, the seasonal change in temperature and salinity was ~1˚C and ~0.1 (psu). Linzmeier et al. [29] presented a composite profile of water column temperature near New Caledonia based on World Ocean Atlas data [41] and showed similar seasonal variation of ~4˚C in the upper 50 m and <1˚C in deeper water. Seasonal variation of surface water $\delta^{18}O$ at New Caledonia has been estimated from seasonal SST and surface salinity variation to be 0.16‰ VPDB [42] and is presumably less in deeper waters. Thus, seasonal fluctuations in temperature and salinity (and $\delta^{18}O_{water}$) at the study site are likely restricted to surface and near-surface water (<50 m). Inasmuch as *N. macromphalus* generally lives below depths of 50 m [28], variations in the upper 50 m are less significant.

**Habitat depth of *Nautilus macromphalus* versus other species of *Nautilus*.** As noted above, few studies of modern nautilus combine C and O isotope measurements of the shell with information about the temperature and isotope composition of the water column. Oba et al. [16] presented results of $\delta^{18}O$ in the septa and outer shell wall of *Nautilus pompilius* collected in Fiji and the Philippines, along with corresponding $\delta^{18}O$ measurements of the water. Temperature profiles of the same stations were included for the Philippines in Hayasaka et al. [43] and for Fiji in Hayasaka et al. [44]. We draw on these data and use our approach to calculate the $\delta^{18}O$ of precipitated aragonite as a function of depth in the water column for the two sites. We note that Oba et al. [16] used ad hoc temperature equations to calculate the

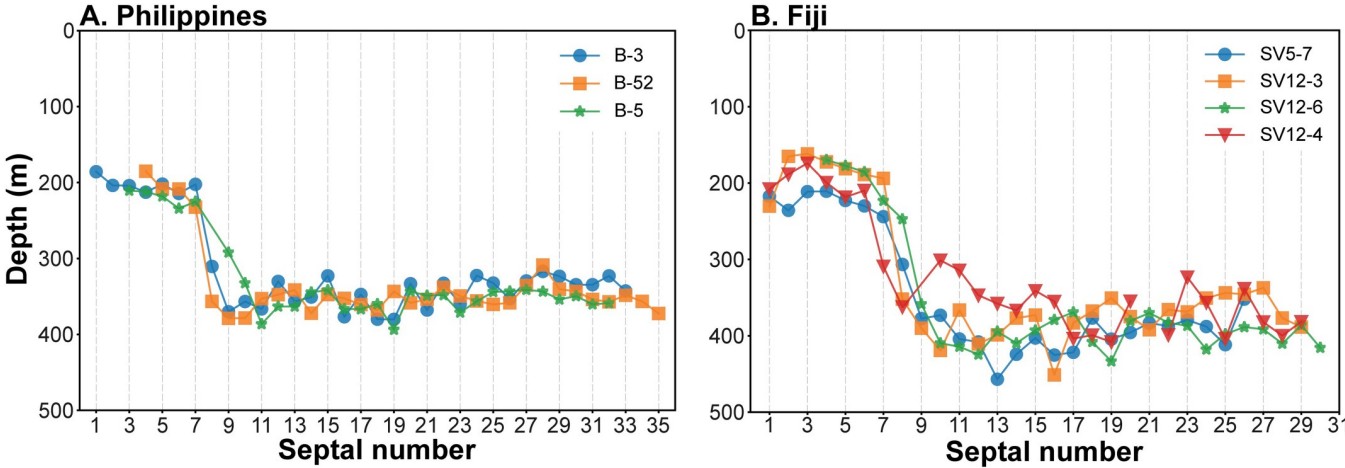

**Fig 9. Reconstructed habitat depth of *Nautilus pompilius* from the Philippines and Fiji.** Depth vs. septal number in the Philippines (A) and Fiji (B), based on the δ18Oar vs. depth relationships. Data from Oba et al. [16] and Hayasaka et al. [43, 44].

temperatures of precipitation. This was pointed out by Landman et al. [23], and here we apply Eq 1 to a reevaluate the data of Oba et al. [16]. The results (Fig 9A and 9B) show that pre-hatching septa were deposited in water depths of ~170–230 m (~20–22˚C) in Fiji and 185–230 m (~20–22˚C) in the Philippines. Post-hatching septa were deposited at depths of 300–430 m (~13–16˚C) in Fiji and 310–400 m (~14–16˚C) in the Philippines. Our average depth for pre-hatching septa in *N. macromphalus* from New Caledonia (~140 m at ~21˚C) is slightly shallower than the depths in the Philippines and Fiji. On the other hand, the average depth of post-hatching septa in New Caledonia (370 m) is similar to that of *N. pompilius* from both the Philippines and Fiji. Taken at face value, the data suggest that *N. pompilius* in Fiji and the Philippines hatch at slightly deeper water than in New Caledonia, but that adults live at or range over similar depths (temperatures) at all three sites. However, the temperature recorded in the pre-hatching stage at all three sites is the same (~20–22˚C), suggesting that water temperature rather than depth is the controlling factor in egg-laying and incubation.

**Carbon isotopes in modern nautilus and fossil nautilids: Identifying carbon sources.** The carbon isotope signature of molluscan aragonite is more difficult to interpret than that of oxygen in that it is only very weakly dependent on temperature [20, 23, 37]. It is a function of the isotopic composition of carbon incorporated via the metabolism of the animal as well as that of carbon in the dissolved inorganic carbon (DIC) reservoir [17]. This dependence may be expressed as:

$$\delta^{13}C_{ar} = \varepsilon + C_{meta} \times \delta^{13}C_{meta} + (1 - C_{meta}) \times \delta^{13}C_{DIC} \qquad (3)$$

where $\varepsilon$ is the $\delta^{13}C$ fractionation between aragonite and DIC (+2.7 ± 0.6‰, dominated by $HCO_3^-$ [45]), $C_{meta}$ is the fraction of metabolic carbon incorporated into the shell, and $\delta^{13}C_{ar, meta, DIC}$ are the $\delta^{13}C$ values corresponding to the shell, metabolic carbon, and DIC, respectively. In the present study, we use literature values of the $\delta^{13}C$ of the DIC to interpret the fraction of metabolic carbon as recorded in the $\delta^{13}C$ of the septa. The $\delta^{13}C$ of oceanic DIC has been mapped broadly but has changed with time owing to the influence of increasing inputs of fossil fuel $CO_2$ (with low $\delta^{13}C$) into the atmosphere, surface ocean, and upper water column. Measurements of $\delta^{13}C_{DIC}$ in the Pacific show these changes, documented by Ko et al. [35] along a meridional transect using snapshots based on sampling in 1994 and 2008. At the latitude of New Caledonia, the $\delta^{13}C_{DIC}$ in 2008 was ~1.2‰ in the upper ~100 m, decreasing to

~0.8‰ at 400 m [35]. In applying Eq 3 to our nautilus $\delta^{13}$C data, we use $\delta^{13}C_{DIC}$ = 1.2‰ for septa formed pre-hatching (depth ~140 m estimated from $\delta^{18}$O) and 0.8‰ for septa formed post-hatching. The value of $\delta^{13}C_{meta}$ is not known precisely, but Crocker et al. [46] reported measurements of $\delta^{13}$C for siphuncular organic material in six wild-collected specimens of *Nautilus macromphalus* from New Caledonia. Data were not tabulated in Crocker et al. [46], but interpolating values from their plot of $\delta^{13}$C vs. septal number shows that in individual specimens and in the six specimens taken as a whole, there was little difference between the siphuncular material formed pre-hatching (overall average ± 1sd = -16.6 ± 2.2‰ VPDB) and post-hatching (overall average ± 1sd = -16.5 ± 1.4‰ VPDB). The overall average ± 1sd for all material is -16.6 ± 1.7‰ VPDB. This result is comparable to that of Pape [47], who measured -17.4‰ on siphuncular material from a specimen of *Nautilus pompilius* from the Philippines. Given that the values are not significantly different pre- and post-hatching, we use an overall average value (±1sd) of -17‰ ± 2‰ VPDB for $\delta^{13}C_{meta}$ to calculate values of $C_{meta}$ using Eq 3 (Fig 10).

The pattern of change in septal $\delta^{13}$C observed in both specimens of *Nautilus macromphalus* shows higher values of $C_{meta}$ in septa formed pre-hatching (septa 1–7) compared with those formed post- hatching: averages ± 1sd of 21.8 ± 2.5% vs. 14.2 ± 2.4% in AMNH 132423, respectively (Fig 10; S1 Table). Both specimens show an increase in $C_{meta}$ to 24–27% in septa 9 and 10. We interpret this to result from an accelerated rate of growth shortly after hatching. Our results are comparable to those of Chung et al. [18] who calculated $C_{meta}$ values for specimens of nautilus based on previously published data on septal and DIC $\delta^{13}$C values. They calculated that $C_{meta}$ decreased from ~30% to ~10% through ontogeny. The average values of $C_{meta}$ for post-hatching nautilus septa are comparable to the low values, generally less than 10%, observed in other marine mollusks such as bivalves and gastropods [8]. In contrast, values of $C_{meta}$ for another group of shelled cephalopods, extinct ammonites, are estimated to be considerably greater than those of modern nautilids. For example, Tobin and Ward [48] compared

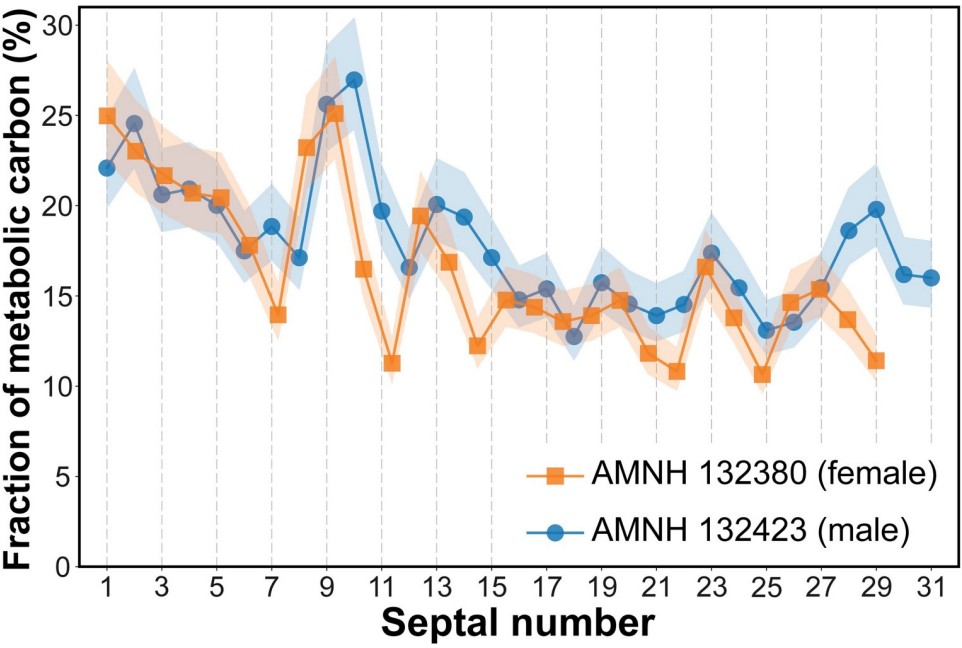

**Fig 10. Reconstructed fraction of metabolic carbon in *Nautilus macromphalus*.** Shaded areas indicate the estimated uncertainty in $C_{meta}$ given the average ±1sd of $\delta^{13}C_{meta}$ (-17 ± 2‰).

$\delta^{13}$C in bivalve and ammonite shells from the Lopez de Bertadano Formation of Seymour Island (Antarctica) spanning the K/Pg boundary. They used the $\delta^{13}$C of bivalves to calculate the $\delta^{13}$C of the DIC (assuming 10% metabolic C in the bivalves), and then used Eq 3 to calculate the average $C_{meta}$ values ranging from 31.8–36.8% in ammonites. Landman et al. [49] used the detailed $\delta^{13}$C data of Sessa et al. [50] measured on foraminiferas, gastropods, bivalves, and ammonites from the Upper Cretaceous Owl Creek Formation (Mississippi, USA) to calculate the $C_{meta}$ of 29% for *Eubaculites* sp.

Given that modern nautilids appear to incorporate less metabolic carbon in their shells than did the extinct ammonites, it is interesting to consider whether the $\delta^{13}$C of extinct shelled nautilids can be used to estimate the $\delta^{13}$C of the DIC in which the nautilids lived. We consider two examples, both involving *Eutrephoceras dekayi*, an extinct nautilid often compared with modern nautilus [14, 33, 51]. In both cases, preservation of the shell microstructure of the samples was assessed using the Preservation Index of Cochran et al. [10] and determined to be excellent (PI = 5). One sample of *E. dekayi* from the Owl Creek Formation included in Sessa et al. [50] showed a value of $\delta^{13}$C of 0.76‰. Assuming $C_{meta}$ = ~-15% and $\delta^{13}C_{meta}$ = ~-17‰, application of Eq 3 yields a value of $\delta^{13}C_{DIC}$ of 0.7‰. This compares favorably with the $\delta^{13}$C values of benthic foraminifera at the site (0.8–0.9‰), which predominantly reflect the $\delta^{13}$C of the DIC.

Another example is from the work of Cochran et al. [52], which documents the $\delta^{13}$C of a shell of *Eutrephoceras dekayi* collected at a fossil methane seep deposit in the Upper Cretaceous *Baculites compressus* ammonite Zone in the Pierre Shale of South Dakota. The outer shell of the specimen was sampled sequentially to determine $\delta^{18}$O and $\delta^{13}$C through ontogeny. Values of $\delta^{13}$C varied from 0.15‰ to -0.32‰, generally decreasing through ontogeny. The corresponding calculated values of $\delta^{13}C_{DIC}$ range from -0.6‰ to -0.0‰ (average -0.24‰). The lower calculated values of $\delta^{13}C_{DIC}$ at this site are consistent with the influence of the anaerobic oxidation of methane with low $\delta^{13}$C impacting the DIC reservoir in near-surface sediments. The presence of seep fluids in the immediate overlying water column at the site is supported by $^{87}$Sr/$^{86}$Sr ratios in the shells of ammonites and nautilids that differ from coeval seawater values [52], as well as by patterns of $\delta^{13}$C in the shells of *B. compressus* and other ammonites [13, 53]. Future work is needed to determine whether the $\delta^{13}$C of fossil nautilid shells can reasonably be used as a proxy for $\delta^{13}$C of paleo-DIC. This may permit the reconstruction of paleoenvironments over long geological periods ranging from the Carboniferous [54] to the Miocene [55].

## Morphology and isotope signals

Elucidating the ecology of fossil cephalopods, including their habitat and migratory behavior is of great importance to better understand the mechanisms of their evolution and extinction. As mentioned, however, these aspects can only be studied based on indirect evidence. In both fossil and modern nautilids, the hard part (i.e., conch) is the most accessible material and so the morphology of the conch is the focus of most studies. Stable isotopes are also a useful tool to reconstruct ecology, particularly habitat, when the original aragonitic shell is preserved. However, the conch of cephalopods is susceptible to diagenetic changes and, as a result, the original shell material is often altered. If there is a link between isotope signals and morphological changes, then cephalopods that do not preserve their original shells can provide valuable information.

Our results reveal that changes in morphological parameters are apparent at some ontogenetic stages. The most conspicuous morphological changes occur at the point of hatching (i.e., a conch diameter of about 30 mm corresponding to septum 8) as expressed in septal spacing,

whorl expansion rate (WER), whorl width index (WWI), and siphuncle position index (SPI) (Fig 6). These morphological changes most likely occur in all species of modern nautilids according to the results of recent studies [25, 56]. The point of hatching also coincides with changes in $\delta^{18}O$ and $\delta^{13}C$, as observed in the two specimens of *Nautilus macromphalus* we studied.

Another conspicuous morphological change occurs in late ontogeny—known as the morphogenetic count down [25, 57]. A direct comparison of the patterns of septal spacing and $\delta^{18}O$ reveals that the onset of the morphogenetic countdown, expressed as an approximation of the last two septa, approximately coincides with a decrease in the value of $\delta^{18}O$ in both specimens, indicating a preference for more shallow water (Figs 5B, 6A, and 6B) [37]. Other morphological parameters (WER and WWI) are more difficult to compare with the changes in isotopes because of the uncertainty of the conch diameter at which a particular septum formed. As mentioned in the Materials and Methods section, although we estimated the body chamber length at the time each septum formed using the length of the body chamber as an adult, the estimated position of the aperture may not be exact. Bearing that in mind, the inferred conch diameters at which each septum formed are presented in S1 Table. The estimated conch diameters at which each septum formed are presented in S1 Table. This allows for a comparison through ontogeny between WER, WWI, on the one hand, and the isotope values, on the other hand. It is notable that the change in WER and WWI, corresponding to septum 29, approximately coincides with a change in $\delta^{18}O$ and $\delta^{13}C$ (Table 1).

A conspicuous change in $\delta^{13}C$ occurs at septum 13. Although there are no obvious changes in most conch parameters at septum 13, the change in $\delta^{13}C$ appears to coincide with the onset of a decreasing trend of WER (Fig 6E and 6F). It is not clear whether this is a mere coincidence or a biological/ecological signal. In addition, we note several other minor changes in morphological parameters, namely in WER and WWI (Fig 6). The changes in $\delta^{18}O$ and $\delta^{13}C$ that correspond to the septal numbers at which these morphological changes occur are difficult to evaluate. We calculated Spearman's rank correlation coefficients between SPI, septal spacing, and isotope signals (Table 2). The results indicate that the values of $\delta^{18}O$ and $\delta^{13}C$ are not correlated to SPI and septal spacing with one exception—septal spacing in AMNH 132423. Given these results, we conclude that the morphological changes are not clearly reflected in the values of $\delta^{18}O$ and $\delta^{13}C$ during middle ontogeny [i.e., after hatching until the onset of morphogenetic countdown (sensu Seilacher and Gunji [57])]. As demonstrated by previous researchers [29], conventional methods of analyzing $\delta^{18}O$ and $\delta^{13}C$ usually require significant amounts of aragonite for each sample and involve time-averaging (i.e., multiple days/months of growth are averaged), and therefore, may have masked detailed changes in morphology. Highly resolved sampling may shed new light on the relationship between morphology and stable isotopes [29,

**Table 1. Comparison of the onset of maturity and changes in isotope composition of the septa.**

| | Septal number at which a significant change occurs in late ontogeny | |
| --- | --- | --- |
| | AMNH 132380 (female) | AMNH 132423 (male) |
| $\delta^{18}O$ | 28 | 30 |
| $\delta^{13}C$ | 28 | 30 |
| Septal spacing | 29 | 31 |
| Siphuncle position index | 29 | 31 |
| Whorl expansion rate | 29 | 27 |
| Whorl width index | 29 | 29 |

Septal numbers at which a significant change occurs in late ontogeny. The septal numbers are indicated in Fig 6.

**Table 2. Spearman's rank correlation coefficients between morphological parameters septal spacing and siphuncle position index (SPI) and isotopic values ($\delta^{18}$O and $\delta^{13}$C).**

| Correlation coefficient | | $\delta^{13}$C | $\delta^{18}$O |
|---|---|---|---|
| AMNH 132380 | SPI | -0.1971 | 0.0465 |
| | Septal Spacing | -0.2632 | -0.4068 |
| AMNH 132423 | SPI | 0.0714 | -0.0273 |
| | Septal Spacing | -0.4403 | -0.3494 |
| *p*-value | | $\delta^{13}$C | $\delta^{18}$O |
| AMNH 132380 | SPI | 0.4315 | 0.8547 |
| | Septal Spacing | 0.2902 | 0.0938 |
| AMNH 132423 | SPI | 0.7583 | 0.9079 |
| | Septal Spacing | 0.0472* | 0.1210 |

*Statistically significant ($p < 0.05$)

58]. Another potential approach to explore the relationship between morphology and isotope values is specimens with sub-lethal injuries. Shell fracture and repair are relatively common in modern nautilus and are manifested by changes in septal spacing [59, 60]. It is possible that injured specimens also change their migratory behavior pattern to avoid predators, which could be reflected in the isotope signatures. In addition, it is worth mentioning that ammonoids may exhibit a different relationship between morphological and isotope changes than nautilids. As suggested by a previous study [51], ammonites likely responded to changes in environment more rapidly than nautilids due to differences in their respective metabolic rates. Therefore, further investigation with additional specimens of nautilids and ammonoids is valuable.

## Conclusions

We analyzed $\delta^{18}$O and $\delta^{13}$C in the septa of *Nautilus macromphalus* from New Caledonia. We also analyzed water samples from two sites near where the nautilus specimens were collected to determine the temperature-salinity-$\delta^{18}$O profiles as a function of depth. We summarize our conclusions:

1. We reconstructed the habitat depth of *N. macromphalus* from New Caledonia as ~140 m (~21°C) pre-hatching and ~370 m (~15°C) post-hatching. The pre-hatching habitat depth of *N. macromphalus* differs slightly from that for *N. pompilius* from the Philippines (185–230 m) and Fiji (170–230 m), as recalculated from the data of Oba et al. [16]. Nevertheless, the pre-hatching temperature seems to be the same in all three groups (20–22°C). The post-hatching depth is also similar among the three sites. In addition, there is no difference between the female and male of *N. macromphalus* from New Caledonia. At maturity, both specimens of this species show a change to lower values of $\delta^{18}$O, reflecting migration to shallower (warmer) water, possibly related to mating or the search for egg-laying sites. However, as emphasized in the Background section, these estimates represent averages and do not necessarily imply a single preferred depth, but instead are integrated over time and space.

2. Using published data and our results, we estimate that the average fraction of metabolic carbon in *N. macromphalus* is ~21% before hatching and ~15% after hatching. These values are significantly lower than those inferred for fossil ammonites. In addition, we suggest that our results about metabolic carbon may be useful in estimating the $\delta^{13}$C of the DIC in ancient oceans.

3. We carried out morphometrics for *N. macromphalus* using computed tomography to calculate four conch parameters (septal spacing, septal position index, whorl expansion rate, and whorl width index). Changes in isotope values ($\delta^{13}$C and $\delta^{18}$O) and changes in morphology coincide at the point of hatching and at the onset of maturity. Although both the conch parameters and isotope values fluctuate during middle ontogeny, there does not seem to be a clear correlation between them. This suggests that minor changes in the environment barring injuries, are not reflected in the morphology of the conch after hatching until the onset of the morphogenetic countdown. Further investigation is needed to discern more precisely the exact relationship between isotope signals and morphological changes in ectocochleate cephalopods.

## Supporting information

**S1 Table. Raw data of $\delta^{18}$O, $\delta^{13}$C, and morphological parameters and calculated fraction of metabolic carbon.**
(XLSX)

## Acknowledgments

We thank Morgan Chase (AMMH) and Andrew Smith (Formerly AMNH) for CT-scanning the specimens. Steve Thurston (AMNH) is thanked for photographing the specimens. Mariah Slovacek (Formerly AMNH) and Anastasia Rashkova (AMNH) are thanked for help in processing the specimens. Royal Mapes (Ohio) is thanked for donating the specimens to AMNH. Dyke Andreasen Andreasen, Colin Carney (University of California, Santa Cruz), and Lora Wingate (University of Michigan) are thanked for help and patience in the isotope analyses of samples. AT thanks Kozue Nishida (University of Tsukuba) for fruitful discussions. We would like to thank an anonymous reviewer and Benjamin J. Linzmeier (University of South Alabama) for their constructive comments.

## Author Contributions

**Conceptualization:** Amane Tajika, Neil H. Landman, J. Kirk Cochran.

**Data curation:** Amane Tajika, Neil H. Landman, J. Kirk Cochran.

**Formal analysis:** Amane Tajika.

**Funding acquisition:** Amane Tajika, Neil H. Landman, J. Kirk Cochran.

**Investigation:** Amane Tajika, Neil H. Landman, J. Kirk Cochran, Claire Goiran, Aubert Le Bouteiller.

**Methodology:** Amane Tajika, Neil H. Landman, J. Kirk Cochran, Claire Goiran, Aubert Le Bouteiller.

**Writing – original draft:** Amane Tajika, Neil H. Landman, J. Kirk Cochran.

**Writing – review & editing:** Amane Tajika, Neil H. Landman, J. Kirk Cochran, Claire Goiran, Aubert Le Bouteiller.

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
