## [Decision Letter · Decision Letter 0]

3 Feb 2022

PONE-D-21-35042Refining the habitat of Nautilus macromphalus based on knowledge of the O and C isotope composition and temperature of the water column in New CaledoniaPLOS ONE

Dear Dr. Tajika,

Thank you for submitting your manuscript to PLOS ONE. After careful consideration, we feel that it has merit but does not fully meet PLOS ONE’s publication criteria as it currently stands. Therefore, we invite you to submit a revised version of the manuscript that addresses the points raised during the review process. Please submit your revised manuscript by Mar 20 2022 11:59PM. If you will need more time than this to complete your revisions, please reply to this message or contact the journal office at plosone@plos.org. Please include the following items when submitting your revised manuscript:A rebuttal letter that responds to each point raised by the academic editor and reviewer(s). You should upload this letter as a separate file labeled 'Response to Reviewers'.A marked-up copy of your manuscript that highlights changes made to the original version. You should upload this as a separate file labeled 'Revised Manuscript with Track Changes'.An unmarked version of your revised paper without tracked changes. You should upload this as a separate file labeled 'Manuscript'.

We look forward to receiving your revised manuscript.

Kind regards,

Geerat J. Vermeij

Academic Editor

PLOS ONE

https://journals.plos.org/plosone/s/file?id=ba62/PLOSOne_formatting_sample_title_authors_affiliations.pdf"

“AT was supported by a Grant-in-Aid for JSPS Research Fellow, Grant-in-Aid for Young Scientists (grant nrs. 20J00376 and 21K14028). NHL and JKC were supported by the Norman D. Newell Fund (AMNH).”

 “AT was supported by a Grant-in-Aid for JSPS Research Fellow, Grant-in-Aid for Young Scientists (grant nrs. 20J00376 and 21K14028). NHL and JKC were supported by the Norman D. Newell Fund (AMNH). The funders had no role in study design, data collection and analysis, decision to publish, or preparation of the manuscript.”

4. We note that [Figure 1] in your submission contain [map/satellite] images which may be copyrighted. All PLOS content is published under the Creative Commons Attribution License (CC BY 4.0), which means that the manuscript, images, and Supporting Information files will be freely available online, and any third party is permitted to access, download, copy, distribute, and use these materials in any way, even commercially, with proper attribution. For these reasons, we cannot publish previously copyrighted maps or satellite images created using proprietary data, such as Google software (Google Maps, Street View, and Earth). For more information, see our copyright guidelines: http://journals.plos.org/plosone/s/licenses-and-copyright.

 a. You may seek permission from the original copyright holder of Figure(s) [#] to publish the content specifically under the CC BY 4.0 license. 

Additional Editor Comments:

The two reviewers raise truly important issues regarding your study. It may prove very difficult to address yourselves to all the problems, but at the very least you should acknowledge the problems even if you can't fix them or solve them. On a personal level, I have been a skeptic of isotopic studies like this for some of the reasons cited by Reviewer 2, especially the question when shell secretion actually takes place. If you do resubmit this paper, I shall have it critically reviewed again.

Reviewers' comments:

Reviewer's Responses to Questions

**Comments to the Author**

1. Is the manuscript technically sound, and do the data support the conclusions?

Reviewer #1: Yes

Reviewer #2: Partly

2. Has the statistical analysis been performed appropriately and rigorously? 

Reviewer #1: N/A

Reviewer #2: Yes

3. Have the authors made all data underlying the findings in their manuscript fully available?

Reviewer #1: Yes

Reviewer #2: Yes

4. Is the manuscript presented in an intelligible fashion and written in standard English?

Reviewer #1: Yes

Reviewer #2: Yes

5. Review Comments to the Author

Reviewer #1: Dr. Tajika and colleagues present an excellent study that combines the investigation of morphological variation and stable isotope variability through ontogeny within Nautilus macromphalus from New Caledonia. These data extensively explore the depth-metabolism-morphology covariation in an extant externally shelled cephalopod. The manuscript and supporting information are well presented and clear. I have a few general comments to help improve the manuscript and some more specific suggestions for sentences and figure changes.

I believe that this study requires a short ethics statement given the locally protected status of Nautilus macromphalus and the PLoS expectations for ethics statements related to cephalopods (https://journals.plos.org/plosone/s/animal-research). Mostly a statement on the timing of collection and the purpose of collection would suffice. I should reiterate that I believe the work these authors have done has been conducted ethically, but given the journal requirements, this statement should be added.

I would like to see slightly more nuanced phrasing related to the depth estimates throughout the manuscript. Later in the text it is clearly stated that these estimates are averages given septal formation duration, but it would be helpful to state early-on that “depth estimates average X days growth”. There is also an implicit assumption that septal formation rate is constant at all depths, which is probably fine given inferences from apertural growth and variance observed in subsampling septa by other authors. In addition, the time averaged within each septa likely varies with growth, but the possible depth range (or swimming velocity) also likely scales with size. Changing the time-at-depth distribution coupled to the time averaging could change the observed mean depth habitat in a free-swimming organism like Nautilus especially during the morphogenic count down (Linzmeier, 2019).

One potential area of improvement would be to do a “time averaging” to the morphological parameters and compare these averaged values to the resultant δ18O and δ13C. It is unclear to me from the methods description if the quantification is done at a point along the shell or an average for a region. Currently the comparison between the isotope and morphological data is done broadly and descriptively but quantitative comparison (e.g. regression results) could be explored. For this to be fruitful, major changes in morphology and isotope ratios at the hatching and morphogenetic count down would likely need to be trimmed from the regression. Remaining morphological parameters and isotope results could show a statistically significant, but low R2 regression. Doing this isn’t entirely necessary for publication but could show more subtle changes than currently described in the manuscript.

I would be interested in seeing some speculation about potential covariation in morphology and δ18O within shells that exhibit sub-lethal injuries. I would expect there to be a depth change to either avoid predation during healing or to compensate for hydrostatic pressure during repair. Some notes on what ammonoids would be most likely to show morphology-geochemistry covariation would also be useful and likely based on morphological variance.

Overall, this is a well-presented manuscript and I look forward to seeing it published. The data and methods presented are useful and provide a good framework for exploring the paleobiology of cephalopods. If the authors have any questions or would like any clarification on my comments, do not hesitate to contact me.

- Ben Linzmeier (blinzmeier@southalabama.edu)

Line by line comments (No line numbers so copied text is first followed by indented comment):

Using δ13C of the shells and published data

I would suggest rephrasing to “Using δ13C of shell carbonate and published data on metabolic carbon” or something similar.

morphological parameters and the changes in δ13C and δ18O during ontogeny do not coincide except at hatching and at the onset of maturity

It may be useful to relate this to an interpretative conclusion.

a complex picture of their habitat

May be useful to rephrase to “complex picture of their behavior within their habitat.”

This is due to the fact that trapping for nautilus and collecting water samples at the same time is a difficult proposition.

I think this is only half the story because the analysis of δ18O of water and δ13C of DIC are both more difficult in some ways than the measurement of δ18O and δ13C of carbonates using gas-source mass spectrometry.

This is because the δ13C of the shell is both a function of carbon incorporated via the metabolism of the animal as well as from a function of the dissolved inorganic carbon (DIC).

Incorporate some references in this paragraph. You have them cited later in the paper, but putting them here would be useful, too.

Highly resolved morphological examination of nautilid conchs was difficult in the past but is now possible owing to the advancement of tomographic methods.

Add some more specificity here about the scale and type of morphological change that can now be resolved.

ground to a powder before isotope measurement

Can you add more specifics here? I assume mortar and pestle for grinding rather than using a rotary tool.

Water samples were also collected in June, 2003

Remove “also” here.

were also analyzed for δ18O of the water and δ13C

Remove “also” here.

The CT-scans obtained were used to measure the following morphological parameters (Fig. 2): conch diameter (dm), whorl width (ww), whorl height (wh), distance between the ventral edge of the siphuncle and the ventral edge

Being a bit more specific here would help bolster your discussion of the influence of time averaging on the comparison of morphology to isotope composition. From what is here, I do not know if each measurement is discrete or if you have a continuous collection of data (I assume discrete given the figures).

However, we note that nautilid septa represent time-averaged growth increments, which increase over ontogeny [19, 30, 31].

Clarify this statement a bit more. Do you mean the time averaged within each septum increases though ontogeny? I assume that is what you mean.

Thus, the calculated depths and temperatures based on analyses of the entire septal thicknesses represent temporal averages.

Add a mention of the approximate time averaged within a single analysis.

Detailed measurements by SIMS of δ18O in the outer shell of

It would be useful to write out “Secondary Ion Mass Spectrometry” for this acronym somewhere within the paper.

Given that our water samples were collected at one time, the septa we analyzed did not form in the exact seawater we sampled.

There are also concerns about along-reef movement. Some telemetry datasets show this information. In addition, the δ18O differences between the collection sites you show should be discussed somewhere. The spatial heterogeneity of the δ18O should also be addressed slightly. I assume submarine groundwater discharge may be important?

Previous studies documented temporal and seasonal variation of seawater temperature and chemistry at/near New Caledonia.

Data from the World Ocean Atlas (Locarnini et al., 2013) also shows some of this seasonality, although with less spatial resolution.

The value of δ13Cmeta is not known precisely, but Crocker et al. [39] and Pape [40] reported measurements of -17‰ for siphuncular organic material in nautilus, and we use that value in estimating Cmeta using eqn. 2.

Given the paucity of data, it may be useful to report what the range in δ13Cmeta would need to be in order to produce no metabolic change in the results of equation 2.

Both specimens also show an increase to 24–25% in Cmeta in septa 9 and 10.

It is likely important to consider variation in δ13Cdiet coinciding with the loss of the egg-yolk source of carbon and transition to other foods. From what I understand, you are assuming a constant δ13C of DIC and diet to calculate the metabolic rate.

They calculated that Cmeta decreased from ~30% to ~10% through ontogeny. The average values of Cmeta for post-hatching nautilus septa are comparable to the low values, generally less than 10%, observed in other marine mollusks such as bivalves and gastropods [8].

Chung et al., 2021 also mention that “the Cresp values before septum 10 are biased due to uncertain δ13Cdiet and δ13CDIC values, rendering the evaluation of ontogenetic variation in metabolism in the egg stage difficult.” Some discussion of pre-hatching δ13Cdiet and δ13CDIC may relate to the permeability of the egg case or changes in the δ13C of the diet given large contrasts between the yolk and post-hatching diet as explored in ammonites (Linzmeier et al., 2018).

Future work is needed to determine whether the δ13C of fossil nautilid shells can reasonably be used as a proxy for δ13C of paleo-DIC.

It may be good to point out the potential time intervals for some of the preserved nautilid carbonates here. I expect you may want to refer to the Buckhorn Asphalt?

Indeed, many studies of extinct cephalopods examine conch morphology.

Include some more references here.

Our results reveal that the changes in δ18O and δ13C also coincide with

Potentially reduce the use of “also” in this paragraph and potentially throughout the manuscript. It is not terribly distracting, but potentially unnecessary in some places.

We estimated the body chamber length at the time each septum formed using the length of the body chamber at adult.

Ohno et al., 2014 use a slightly more complicated approach to compare the δ18O of septa to shell wall and increase the size of the body chamber during growth given the results of a regression across many shells. It is unlikely to change the results with this adjustment, but may be worth mentioning here. In addition, the body chamber length varies periodically with the septal formation cycle, reaching a maximum size immediately before the precipitation of the septa during mural ridge formation which adds additional complexity to the septum-shell wall correlation (Ward et al., 1981). It’s a sticky problem to solve, but you could potentially do some sort of moving window/expanding window averaging to test sensitivity in the future.

Another conspicuous morphological change occurs in late ontogeny—known as the morphogenetic count down

It is probably worth citing (Zakharov et al., 2006) somewhere in this paragraph because they report similar changes in the δ18O of Nautilus during the morphogenetic count down.

resolved sampling may shed new light on the relationship between morphology and stable isotopes, although such methods also suffer from time-averaging to some degree

The math of the time averaging combined with swimming is incorporated into Linzmeier, 2019. For instance, Figure 5 in that paper shows the expected reduction in variance associated with averaging 23 days/analysis compared to random replicate sampling with 2 days/analysis. So the septal δ18O minimize expected depth variance even more than what is modeled. Differing vertical swimming velocities through ontogeny could also further interact with the time averaging.

Figure Comments.

Figure 1.

It may be useful to outline the possible Nautilus habitat around New Caledonia using implosion depth as the limiting factor.

Figure 3.

It would be useful to add curves from the World Ocean Atlas data product to show how these data relate to what are available there (including the seasonal component of variability). In addition, it would be useful to plot these with the full possible water depth distribution of Nautilus macromphalus linked to the Y axis. You could have panel C as its own figure and combine A and B with Figure 4 to have slightly different aspect ratios for the figures.

Figure 4.

I think the mean line is not necessarily appropriate for this figure given the number of levels that the mean is only of one analysis. It could either be dropped or using LOESS, or polynomial fit may be more appropriate to summarize the data given some depths with single observations.

Figure 5.

Noting the analytical precision on these figures would be useful.

Figure 6.

I find the inset plots slightly distracting. Given the body of the paper it may be useful to separate these into an additional figure that could be focused on only post-hatching variability. It would leave a fair amount of blank-space on this figure, but that could be adjusted by using a log y axis?

Figure 7.

I would like to see the R2, p-value and a 95% confidence interval around the regression.

Figure 8.

It would be useful to incorporate some sort of error estimation around these lines to illustrate the combined effects of error from the δ18O analyses and regression models.

Figure 9.

Because you compare the depths of the N. macromphalus to these, I would like to see subtle grey lines showing these data as a point of reference.

Figure 10.

Adding an annotation based on comments from Chung et al., 2021, about uncertainty in the pre-hatching metabolic rate given uncertainty in diet and DIC would be useful. The assumptions of the calculation may be violated for these data.

Works cited

Linzmeier, B.J., 2019, Refining the interpretation of oxygen isotope variability in free-swimming organisms: Swiss Journal of Palaeontology, v. 138, p. 109–121, doi:10.1007/s13358-019-00187-3.

Linzmeier, B.J., Landman, N.H., Peters, S.E., Kozdon, R., Kitajima, K., and Valley, J.W., 2018, Ion microprobe–measured stable isotope evidence for ammonite habitat and life mode during early ontogeny: Paleobiology, v. 44, p. 684–708, doi:10.1017/pab.2018.21.

Locarnini, R.A. et al., 2013, World Ocean Atlas 2013, Volume 1: Temperature. (S. Levitus, Ed.): NOAA Atlas NESDIS 73, 40 p.

Ward, P., Greenwald, L., and Magnier, Y., 1981, The Chamber Formation Cycle in Nautilus macromphalus: Paleobiology, v. 7, p. 481–493.

Zakharov, Y., Shigeta, Y., Smyshlyaeva, O., Popov, A., and Ignatiev, A., 2006, Relationship between δ13C and δ18O values of the Recent Nautilus and brachiopod shells in the wild and the problem of reconstruction of fossil cephalopod habitat: Geosciences Journal, v. 10, p. 331–345, doi:10.1007/BF02910374.

Reviewer #2: Experiments, statistics, and other analyses are performed to a high technical standard and are described in sufficient detail?

Partly.

There are a number of important details explained in the discussion section that I would like to see explained earlier, particularly details relevant to making sense of the methods. For example, the methods state that the authors did not do high resolution isotopic analyses of growth increments but rather low spatial resolution sampling of septa. This raises all sorts of questions that should not be left until the end, such as whether this scale is appropriate to the question. That should be clear to the reader in the methods and not something to save for the discussion. How much time is represented by a single septum? Nautiloids are also known to migrate vertically and horizontally daily and should experience a range of environmental conditions. Do nautiloids grow new shell only when at certain preferred depths? What if they live most of the time in deeper waters but form new shell when they are more metabolically active in shallower waters? How do the authors know this isn’t the case? Or is the isotopic record of the nautilus shell averaged across environments (i.e., if they grow more or less continuously)? If the isotopic proxy for environment is averaged and not representative of a single environment, is it valid to use the regression equation for depth vs. predicted d18O values? The authors may be right in everything they did, but they do not make a strong enough justification that their approach works based on the methods section alone. The discussion section answers some of this, but not all of it, and the reader should have some confidence the methods are appropriate before they see the results.

There are many questions about the authors’ attempt to analyze d13C of sampled waters. There is no information on how much time elapsed between water sample collection and analysis. Samples can often be stored for some time if they have been treated properly. However, the methods say only that the samples were stored in the dark. This would not have prevented bacterial alteration of DIC and d13CDIC in the water samples. In the discussion, oddly enough, the authors admit they did not dose the samples with HgCl2 as required to stop bacterial activity. I’m puzzled why this didn’t happen in the first place, why it isn’t mentioned in the methods just the discussion, why the analyses were done at all after the early misstep, and why the authors reported the data knowing the data can not be interpreted as d13CDIC.

The authors used literature values of d13CDIC, but, as before, very little is explained in the methods about how were the data selected from Ko et al. (2014). Was a single value used? If so, why one? These answers are provided but only in the discussion. If it’s more efficient to leave the writing as is, fine, but at least refer the reader to “x, y, and z are explained in the discussion.”

The sampling of septa is a low-resolution approach but it seems appropriate to the scale of the question, which is correlation with habitat and morphological change over years.

The estimation of the fraction of metabolic carbon incorporated into the shell from equation 2 (reported from another previously published study – reference should be Crocker 39 and not Pape 40, by the way) relies on several poorly known parameters. One that stands out, besides d13CDIC, is the single value of -17 per mil for organic matter in nautilus. This is one measurement from one species at one point in ontogeny. I looked at the Crocker paper in depth, and I can not figure out why -17 per mil was selected. That paper presented data from 6 nautilus specimens, with different values between and within specimens. Variation within specimens can be as high as 6 per mil. There is no mention of -17 per mil in the text of the paper, so this number will have to be justified.

Conclusions are presented in an appropriate fashion and are supported by the data?

The main finding is that nautilus hatch in shallower, warmer waters and move to deeper depths post-hatching is supported by the use of d18Oshell as a proxy for depth.

Hope is also expressed that even though fossil nautilids can’t often be studied geochemically to reconstruct their reproductive habits/habitats, morphology could serve as a proxy for geochemical changes related to life habits. Other statements about this, however, are inconsistent. In one place, the authors state that changes in geochemistry concide with morphological parameters. Elsewhere, they state that geochemical and morphological changes are “within the range of fluctuation and thus difficult to correlate.” Later, they state that “detectable morphological changes are not clearly reflected” in the geochemistry. Which is it? This message is very muddled, and the significance for future work is not clear.

The article is presented in an intelligible fashion and is written in standard English?

Mostly. The aims of the paper are stated very clearly.

There are a few parts that could be written more clearly. For example, what is “presence of septal approximation”? Define specialized terms for non-specialists.

Equation 2 is missing some parentheses.

There’s a typo in Fig. 6E. One of the septum numbers is missing.

There may be issues with the references. The authors refer to Pape 40, and there is a Pape 40, but they clearly meant Crocker 39. A standard reference check would not catch that. References should be rechecked carefully.

Please show the abbreviated equations in the Fig. 6 caption in the parentheses instead of the acronyms. There’s no point in showing the acronym for the term since the term is already spelled out in the same sentence. However, what a reader may have trouble recalling are the equations for each index. It’s very frustrating as a reviewer or reader to have do a scavenger hunt to be confused about what the terms mean and then have to search the text. In this case, I had to open three windows side by side to figure it out, one for this figure, one for the text definitions of indices, and a third for the figure showing the measurements used in the indices. Ideally, any reader should be able to look at a graph and caption and get it.

Also in Fig. 6, should septal position index be siphuncle position index? The former is not defined in the methods, but the latter is.

Also in Fig 6A and B, the septal spacing graph is hard to compare to the other graphs because x-axis is different. That makes it difficult to look at the graphs an intuitively know whether all the morphological changes are happening at the same point in ontogeny or not.

6. PLOS authors have the option to publish the peer review history of their article (what does this mean?). If published, this will include your full peer review and any attached files.

Reviewer #1: **Yes: **Benjamin J. Linzmeier, PhD

Reviewer #2: No

---

## [Author Response · Author response to Decision Letter 0]

1 Jun 2022

POINT-BY-POINT RESPONSES TO REVIEWS

Comments regarding journal requirements:

“1. Please ensure that your manuscript meets PLOS ONE's style requirements, including those for file naming. The PLOS ONE style templates can be found at

https://journals.plos.org/plosone/s/file?id=ba62/PLOSOne_formatting_sample_title_authors_affiliations.pdf"”

RESPONSE: We corrected the figure citations, the size of headings, and the legend in Table 1. 

“2. Thank you for stating the following in the Funding Section of your manuscript:

“AT was supported by a Grant-in-Aid for JSPS Research Fellow, Grant-in-Aid for Young Scientists (grant nrs. 20J00376 and 21K14028). NHL and JKC were supported by the Norman D. Newell Fund (AMNH).”

 “AT was supported by a Grant-in-Aid for JSPS Research Fellow, Grant-in-Aid for Young Scientists (grant nrs. 20J00376 and 21K14028). NHL and JKC were supported by the Norman D. Newell Fund (AMNH). The funders had no role in study design, data collection and analysis, decision to publish, or preparation of the manuscript.”

Please include your amended statements within your cover letter; we will change the online submission form on your behalf.”

RESPONSE: We removed the funding statement from the manuscript. The current funding statement is good as it is. 

” 3. We note that you have stated that you will provide repository information for your data at acceptance. Should your manuscript be accepted for publication, we will hold it until you provide the relevant accession numbers or DOIs necessary to access your data. If you wish to make changes to your Data Availability statement, please describe these changes in your cover letter and we will update your Data Availability statement to reflect the information you provide.”

RESPONSE: All data are available in S1 Table and thus we would like to update the Data Availability statement.

“4. We note that [Figure 1] in your submission contain [map/satellite] images which may be copyrighted. All PLOS content is published under the Creative Commons Attribution License (CC BY 4.0), which means that the manuscript, images, and Supporting Information files will be freely available online, and any third party is permitted to access, download, copy, distribute, and use these materials in any way, even commercially, with proper attribution. For these reasons, we cannot publish previously copyrighted maps or satellite images created using proprietary data, such as Google software (Google Maps, Street View, and Earth). For more information, see our copyright guidelines: http://journals.plos.org/plosone/s/licenses-and-copyright.

 We require you to either (1) present written permission from the copyright holder to publish these figures specifically under the CC BY 4.0 license, or (2) remove the figures from your submission:”

RESPONSE: We reproduced the map in Fig 1 using OpenStreetMap, which is open data licensed under the Open Data Commons Open Database License (ODbL) by the OpenStreetMap Foundation (OSMF). We added the copy right information to the figure caption.

Comments from Editor: 

“The two reviewers raise truly important issues regarding your study. It may prove very difficult to address yourselves to all the problems, but at the very least you should acknowledge the problems even if you can't fix them or solve them. On a personal level, I have been a skeptic of isotopic studies like this for some of the reasons cited by Reviewer 2, especially the question when shell secretion actually takes place. If you do resubmit this paper, I shall have it critically reviewed again.”

RESPONSE: We revised the manuscript as outlined below. Previous studies have suggested that the secretion of septa is more or less continuous (Ward 1987; Oba et al. 1992; Linzmeier et al. 2016). Although some issues remain, as discussed below, we are explicit in acknowledging and dealing with them. We think that the conclusions we reach are robust. 

Comments from Reviewer #1:

“I believe that this study requires a short ethics statement given the locally protected status of Nautilus macromphalus and the PLoS expectations for ethics statements related to cephalopods (https://journals.plos.org/plosone/s/animal-research). Mostly a statement on the timing of collection and the purpose of collection would suffice. I should reiterate that I believe the work these authors have done has been conducted ethically, but given the journal requirements, this statement should be added.”

RESPONSE: We added Ethics Statement in the Materials and Methods section.

“I would like to see slightly more nuanced phrasing related to the depth estimates throughout the manuscript. Later in the text it is clearly stated that these estimates are averages given septal formation duration, but it would be helpful to state early-on that “depth estimates average X days growth”. There is also an implicit assumption that septal formation rate is constant at all depths, which is probably fine given inferences from apertural growth and variance observed in subsampling septa by other authors. In addition, the time averaged within each septa likely varies with growth, but the possible depth range (or swimming velocity) also likely scales with size. Changing the time-at-depth distribution coupled to the time averaging could change the observed mean depth habitat in a free-swimming organism like Nautilus especially during the morphogenic count down (Linzmeier, 2019).”

RESPONSE: We added more information on the time required to produce each septum. As Landman and Cochran (1987) explained, the time to secrete a septum increases throughout ontogeny. In early ontogeny, a septum may take weeks to be secreted. In later ontogeny, a septum takes months. This pattern is well established, but the actual times are difficult to pin down. The pattern can be inferred from aquarium observations, but the precise timing in nature is more difficult to ascertain.

“One potential area of improvement would be to do a “time averaging” to the morphological parameters and compare these averaged values to the resultant δ18O and δ13C. It is unclear to me from the methods description if the quantification is done at a point along the shell or an average for a region. Currently the comparison between the isotope and morphological data is done broadly and descriptively but quantitative comparison (e.g. regression results) could be explored. For this to be fruitful, major changes in morphology and isotope ratios at the hatching and morphogenetic count down would likely need to be trimmed from the regression. Remaining morphological parameters and isotope results could show a statistically significant, but low R2 regression. Doing this isn’t entirely necessary for publication but could show more subtle changes than currently described in the manuscript.”

RESPONSE: We produced new data on the siphuncle position index (SPI) for each septum and made new graphs. This allowed us to make a direct comparison between septal spacing, SPI, and isotopic values. Using the data, we carried out statistical tests to determine if there is a correlation between them. Accordingly, we produced Table 2 to show the results. It is important to emphasize that the two observed morphological changes indicative of hatching and the onset of maturity are, indeed, manifested in changes in the isotope signature.

“I would be interested in seeing some speculation about potential covariation in morphology and δ18O within shells that exhibit sub-lethal injuries. I would expect there to be a depth change to either avoid predation during healing or to compensate for hydrostatic pressure during repair. Some notes on what ammonoids would be most likely to show morphology-geochemistry covariation would also be useful and likely based on morphological variance.”

RESPONSE: It is a very interesting suggestion, and we added some additional notes on these points at the end of the discussion section. However, the shells that form the basis of this study do not show any significant shell breaks and, therefore, cannot be used to investigate this issue. This topic is the subject of a future study.

“I would suggest rephrasing to “Using δ13C of shell carbonate and published data on metabolic carbon” or something similar.”

RESPONSE: We rephrased the expression, following the suggestion.

“morphological parameters and the changes in δ13C and δ18O during ontogeny do not coincide except at hatching and at the onset of maturity

It may be useful to relate this to an interpretative conclusion.”

RESPONSE: We added some notes on this point to the Conclusions section.

“a complex picture of their habitat

May be useful to rephrase to “complex picture of their behavior within their habitat.”

RESPONSE: We rephrased the expression, following the suggestion.

“This is due to the fact that trapping for nautilus and collecting water samples at the same time is a difficult proposition.

I think this is only half the story because the analysis of δ18O of water and δ13C of DIC are both more difficult in some ways than the measurement of δ18O and δ13C of carbonates using gas-source mass spectrometry.”

RESPONSE: We added this point to the text. 

“This is because the δ13C of the shell is both a function of carbon incorporated via the metabolism of the animal as well as from a function of the dissolved inorganic carbon (DIC).

Incorporate some references in this paragraph. You have them cited later in the paper, but putting them here would be useful, too.”

RESPONSE: We added some references to this part. 

“Highly resolved morphological examination of nautilid conchs was difficult in the past but is now possible owing to the advancement of tomographic methods.

Add some more specificity here about the scale and type of morphological change that can now be resolved.”

RESPONSE: We added some text to specify the morphological changes. 

“ground to a powder before isotope measurement

Can you add more specifics here? I assume mortar and pestle for grinding rather than using a rotary tool.”

RESPONSE: Septal samples were removed using clippers. The samples were then ground down using a mortar and pestle.

“Water samples were also collected in June, 2003

Remove “also” here.

were also analyzed for δ18O of the water and δ13C

Remove “also” here.”

RESPONSE: We removed “also” from the sentences. 

“The CT-scans obtained were used to measure the following morphological parameters (Fig. 2): conch diameter (dm), whorl width (ww), whorl height (wh), distance between the ventral edge of the siphuncle and the ventral edge

Being a bit more specific here would help bolster your discussion of the influence of time averaging on the comparison of morphology to isotope composition. From what is here, I do not know if each measurement is discrete or if you have a continuous collection of data (I assume discrete given the figures).

RESPONSE: We added more details of the data collection. 

However, we note that nautilid septa represent time-averaged growth increments, which increase over ontogeny [19, 30, 31].

Clarify this statement a bit more. Do you mean the time averaged within each septum increases though ontogeny? I assume that is what you mean.”

RESPONE: We rephrased the sentence as suggested. 

“Thus, the calculated depths and temperatures based on analyses of the entire septal thicknesses represent temporal averages.

Add a mention of the approximate time averaged within a single analysis.”

RESPONSE: The exact time of septal formation of N. macromphalus in the wild is not known. We added some estimates from previous studies using aquarium observation and radiometric methods.

“Detailed measurements by SIMS of δ18O in the outer shell of

It would be useful to write out “Secondary Ion Mass Spectrometry” for this acronym somewhere within the paper.”

RESPONSE: We spelled out the acronym.

“Given that our water samples were collected at one time, the septa we analyzed did not form in the exact seawater we sampled.

There are also concerns about along-reef movement. Some telemetry datasets show this information. In addition, the δ18O differences between the collection sites you show should be discussed somewhere. The spatial heterogeneity of the δ18O should also be addressed slightly. I assume submarine groundwater discharge may be important?”

RESPONSE: As discussed in the manuscript, we go over at length the fact that nautilus is a mobile animal. It migrates vertically and horizontally, so isotopic values reflect averages. In addition, the septa themselves are time averaged. We explore the implications of these facts more fully in the manuscript. We added details in the Isotope hydrography section comparing the two water stations. In fact, they are quite similar when �18O is plotted against salinity (plot added to supplemental information). To our knowledge there are no data on submarine groundwater discharge in New Caledonia and it is difficult to discern its possible impact on the �18O profiles.

“Previous studies documented temporal and seasonal variation of seawater temperature and chemistry at/near New Caledonia.

Data from the World Ocean Atlas (Locarnini et al., 2013) also shows some of this seasonality, although with less spatial resolution.”

RESPONSE: We added the reference. However, it is worth noting that seasonal variation is most likely to affect the upper 50 m, which is not the preferred habitat of nautilus.

“The value of δ13Cmeta is not known precisely, but Crocker et al. [39] and Pape [40] reported measurements of -17‰ for siphuncular organic material in nautilus, and we use that value in estimating Cmeta using eqn. 2.

Given the paucity of data, it may be useful to report what the range in δ13Cmeta would need to be in order to produce no metabolic change in the results of equation 2.”

RESPONSE: We have refined the calculation of �13Cmeta using the data of Crocker et al. They measured siphuncular �13C through ontogeny in two wild caught N. macromphalus specimens from New Caledonia. We averaged �13C values from pre- and post-hatching siphuncular material, and the values ae essentially identical in both specimens. This is now discussed in the text and used to refine the calculations of % metabolic carbon in the specimens using eqn. 2.

“Both specimens also show an increase to 24–25% in Cmeta in septa 9 and 10.

It is likely important to consider variation in δ13Cdiet coinciding with the loss of the egg-yolk source of carbon and transition to other foods. From what I understand, you are assuming a constant δ13C of DIC and diet to calculate the metabolic rate.”

RESPONSE: See the response above. The similarity in �13C in pre- and post-hatching siphuncular material suggests that the �13Cdiet doesn’t vary pre- and post-hatching. The yolk and Nautilus prey after hatching essentially represent marine organic matter. What changes is the amount of metabolic carbon incorporated into the shell in the egg and immediately post hatching. The simplest explanation for the data is that the metabolic fraction (and likely growth rate) is high in the early stages of growth in the egg (septa 1-5), deceases toward hatching (septa 7,8) then increases immediately after hatching, before finally reducing to about 15% through the remainder of ontogeny.

“They calculated that Cmeta decreased from ~30% to ~10% through ontogeny. The average values of Cmeta for post-hatching nautilus septa are comparable to the low values, generally less than 10%, observed in other marine mollusks such as bivalves and gastropods [8].

Chung et al., 2021 also mention that “the Cresp values before septum 10 are biased due to uncertain δ13Cdiet and δ13CDIC values, rendering the evaluation of ontogenetic variation in metabolism in the egg stage difficult.” Some discussion of pre-hatching δ13Cdiet and δ13CDIC may relate to the permeability of the egg case or changes in the δ13C of the diet given large contrasts between the yolk and post-hatching diet as explored in ammonites (Linzmeier et al., 2018).”

RESPONSE: See response above re: pre- and post-hatching �13Cdiet. In fact, Chung et al. used the Crocker et al. data to obtain the � 13Cdiet (they get -16.5‰). Their �13CDIC is from Aulclair et al. who give “~0.5‰” with no reference. The �13CDIC may be different in the egg case, but we don’t have the information to address that, and the simplest explanation for the �13C of septa formed in the egg, as noted above, is variation in the fraction of C incorporated into the shell via metabolism. Moreover, our link to fossil nautilids is restricted to post-hatching shell material as a potential indication of ambient �13CDIC.

“Future work is needed to determine whether the δ13C of fossil nautilid shells can reasonably be used as a proxy for δ13C of paleo-DIC.

It may be good to point out the potential time intervals for some of the preserved nautilid carbonates here. I expect you may want to refer to the Buckhorn Asphalt?”

RESPONSE: We added a mention on this point. 

“Our results reveal that the changes in δ18O and δ13C also coincide with

Potentially reduce the use of “also” in this paragraph and potentially throughout the manuscript. It is not terribly distracting, but potentially unnecessary in some places.”

REFERENCES: We also removed most of the “also” occurrences in the manuscript.

“We estimated the body chamber length at the time each septum formed using the length of the body chamber at adult.

Ohno et al., 2014 use a slightly more complicated approach to compare the δ18O of septa to shell wall and increase the size of the body chamber during growth given the results of a regression across many shells. It is unlikely to change the results with this adjustment, but may be worth mentioning here. In addition, the body chamber length varies periodically with the septal formation cycle, reaching a maximum size immediately before the precipitation of the septa during mural ridge formation which adds additional complexity to the septum-shell wall correlation (Ward et al., 1981). It’s a sticky problem to solve, but you could potentially do some sort of moving window/expanding window averaging to test sensitivity in the future.”

RESPONSE: Ohno et al. (2014) suggested a positive correlation between shell diameter and body chamber angle. However, they did not take into account the fact that modern nautilids grow the aperture by 15° on average after the formation of last septum (Collins and Ward 1987). Indeed, their data show that (most likely) adult specimens possess a longer body chamber angle, while the body chamber length of immature specimens seems constant with some variation. Therefore, their method is most likely not appropriate. As for the latter point, it is true that the body chamber length changes during ontogeny, which produces another uncertainty about the estimate of the aperture position. We added this point to the manuscript. 

“Another conspicuous morphological change occurs in late ontogeny—known as the morphogenetic count down

It is probably worth citing (Zakharov et al., 2006) somewhere in this paragraph because they report similar changes in the δ18O of Nautilus during the morphogenetic count down.”

RESPONSE: We added the reference.

“resolved sampling may shed new light on the relationship between morphology and stable isotopes, although such methods also suffer from time-averaging to some degree

The math of the time averaging combined with swimming is incorporated into Linzmeier, 2019. For instance, Figure 5 in that paper shows the expected reduction in variance associated with averaging 23 days/analysis compared to random replicate sampling with 2 days/analysis. So the septal δ18O minimize expected depth variance even more than what is modeled. Differing vertical swimming velocities through ontogeny could also further interact with the time averaging.”

RESPONSE: We added the missing reference. We decided not to add the details to the text because it is slightly off the context. 

“Figure 1.

It may be useful to outline the possible Nautilus habitat around New Caledonia using implosion depth as the limiting factor.”

RESPONSE: It would be interesting to indicate the habitat of Nautilus macromphalus near New Caledonia using implosion depth. However, it is slightly off the topic of this manuscript and is highly speculative due to lack of knowledge about horizontal migration. We also think that excessive information that is not directly related to the main point of this manuscript could be distracting for readers, and thus we decided not to add this information. 

“Figure 3.

It would be useful to add curves from the World Ocean Atlas data product to show how these data relate to what are available there (including the seasonal component of variability). In addition, it would be useful to plot these with the full possible water depth distribution of Nautilus macromphalus linked to the Y axis. You could have panel C as its own figure and combine A and B with Figure 4 to have slightly different aspect ratios for the figures.”

RESPONSE: To our knowledge, no studies have been attempted to detect the possible habitat depth of Nautilus in New Caledonia using ultrasonic telemetry techniques. There are only a few studies that examined δ18O of Nautilus from New Caledonia. But, as mentioned in the manuscript, the information on water chemistry and temperature is missing in those studies. For this reason, we attempt to refine this aspect in our present paper. Therefore, we do not think that it is appropriate to indicate speculative depth in the figures. Also, there is ample evidence (including your own composite plot) suggesting that seasonal variation in temperature is small below 50 m. Finally, the dataset documenting the depth distribution of �18Owater is extremely limited (i.e., virtually non-existent) for this area, and we believe it is preferable to focus on the new data rather than the conventional 0.5‰, assumed to be uniform with depth.

“Figure 4.

I think the mean line is not necessarily appropriate for this figure given the number of levels that the mean is only of one analysis. It could either be dropped or using LOESS, or polynomial fit may be more appropriate to summarize the data given some depths with single observations.”

RESPONSE: As the reviewer points out, there is uncertainty about the temperature in the area/ habitat of Nautilus. We are aware that it may not be completely accurate to use the mean value but using LOESS or polynominal fit does not solve this problem. Thus, we decided to keep the mean value.

“Figure 5.

Noting the analytical precision on these figures would be useful.”

RESPONSE: The analytical precision is cited in the relevant Methods section and is small relative to the measured values. A more significant component of uncertainty is the agreement between laboratories. We have now added a comparison of �13C and �18O of select septa measured by two isotope laboratories and discuss the “uncertainty” produced by these replicate measurements.

“Figure 6.

I find the inset plots slightly distracting. Given the body of the paper it may be useful to separate these into an additional figure that could be focused on only post-hatching variability. It would leave a fair amount of blank-space on this figure, but that could be adjusted by using a log y axis?”

RESPONSE: As the reviewer pointed out, the visibility may have not been high enough. However, we would like to keep the graphs of morphological parameters in a single figure. We found that using a log-transformed axis does not improve the visibility. Thus, following the suggestions of Reviewer #2, we changed the inset plots in a way that both graphs share the same x-axis to increase the visibility. In addition to these changes, we carried out statistical analyses for septal spacing and SPI and produced Table 2 so that readers will not need to visually compare the results. 

“Figure 7.

I would like to see the R2, p-value and a 95% confidence interval around the regression.”

RESPONSE: We updated Fig. 7, following the suggestions.

“Figure 8.

It would be useful to incorporate some sort of error estimation around these lines to illustrate the combined effects of error from the δ18O analyses and regression models.”

RESPONSE: We added shaded areas to Fig 8, which represents the error estimate calculated with the 95% confidence interval of the regression. We also added some new data points of depths calculated with the δ18O values that we obtained from the University of Michigan for inter-lab comparisons. 

“Figure 9.

Because you compare the depths of the N. macromphalus to these, I would like to see subtle grey lines showing these data as a point of reference.”

RESPONSE: We think that doing this will make the data difficult to read for N. pompilius, and we prefer not to add the data that are already provided in Fig. 9. 

“Figure 10.

Adding an annotation based on comments from Chung et al., 2021, about uncertainty in the pre-hatching metabolic rate given uncertainty in diet and DIC would be useful. The assumptions of the calculation may be violated for these data.”

RESPONSE: See the comments above on �13C in diet based on actual siphuncular �13C from wild-caught Nautilus from New Caledonia (Crocker et al.). They show no difference pre- or post-hatching, suggesting that at least the “diet” has the same �13C in both stages. The metabolic rate is certainly different as expressed in the higher fraction of metabolic carbon in the pre-hatching septal �13C. We have added upper and lower bounds on the calculated metabolic fraction to the tables and Fig. 10 based on the average +/- 1sd of the Crocker data.

Comments from Reviewer #2:

“There are a number of important details explained in the discussion section that I would like to see explained earlier, particularly details relevant to making sense of the methods. For example, the methods state that the authors did not do high resolution isotopic analyses of growth increments but rather low spatial resolution sampling of septa. This raises all sorts of questions that should not be left until the end, such as whether this scale is appropriate to the question. That should be clear to the reader in the methods and not something to save for the discussion. How much time is represented by a single septum?” 

RESPONSE: We added this information with references to the Material and Methods section. As noted above in response to reviewer 1, we added more information on the time required to produce each septum. As Landman and Cochran (1987) explained, the time to secrete a septum increases throughout ontogeny. In early ontogeny, a septum may take weeks to be secreted. In later ontogeny, a septum takes months. This pattern is well established, but the actual times are difficult to pin down. The pattern can be inferred from aquarium observations, but the precise timing in nature is more difficult to ascertain.

“Nautiloids are also known to migrate vertically and horizontally daily and should experience a range of environmental conditions. Do nautiloids grow new shell only when at certain preferred depths? What if they live most of the time in deeper waters but form new shell when they are more metabolically active in shallower waters? How do the authors know this isn’t the case? Or is the isotopic record of the nautilus shell averaged across environments (i.e., if they grow more or less continuously)? If the isotopic proxy for environment is averaged and not representative of a single environment, is it valid to use the regression equation for depth vs. predicted d18O values? The authors may be right in everything they did, but they do not make a strong enough justification that their approach works based on the methods section alone. The discussion section answers some of this, but not all of it, and the reader should have some confidence the methods are appropriate before they see the results.”

RESPONSE: It is true that modern nautilids migrate vertically and traverse different environments. Thus, the data presented in our paper are time-averaged. Nevertheless, as mentioned, aquarium experiments and radiometric data suggest that the formation of septa is more or less continuous. We added some additional explanation regarding the formation of septa and time-averaging to the manuscript. Linzmeier et al. (2016) and Oba et al. (1992) documented that shell is secreted continuously or how else could they entrain variation due to migration?

“There are many questions about the authors’ attempt to analyze d13C of sampled waters. There is no information on how much time elapsed between water sample collection and analysis. Samples can often be stored for some time if they have been treated properly. However, the methods say only that the samples were stored in the dark. This would not have prevented bacterial alteration of DIC and d13CDIC in the water samples. In the discussion, oddly enough, the authors admit they did not dose the samples with HgCl2 as required to stop bacterial activity. I’m puzzled why this didn’t happen in the first place, why it isn’t mentioned in the methods just the discussion, why the analyses were done at all after the early misstep, and why the authors reported the data knowing the data can not be interpreted as d13CDIC.”

RESPONSE: The samples were collected by our New Caledonian colleagues, and it was not possible to poison them using HgCl2. Nevertheless, we tried to measure DIC and �13C of DIC but as noted originally in the text, the samples did not reliably preserve the original �13C signature. We have now eliminated those data and subsequent discussion of them from the text and table. 

“The authors used literature values of d13CDIC, but, as before, very little is explained in the methods about how were the data selected from Ko et al. (2014). Was a single value used? If so, why one? These answers are provided but only in the discussion. If it’s more efficient to leave the writing as is, fine, but at least refer the reader to “x, y, and z are explained in the discussion.”

RESPONSE: Re: �13CDIC, we have added a link in the Methods section to the Isotope Hydrography Results and there to the Discussion explaining the choice of �13CDIC values used in the calculation of the fraction of metabolic C incorporated into the septal aragonite. 

“The sampling of septa is a low-resolution approach but it seems appropriate to the scale of the question, which is correlation with habitat and morphological change over years.”

RESPONSE: Yes. Although sampling septa is not a high-resolution approach, we think that it is reasonable to estimate the habitat depth and morphological change. To make things clearer, we added more information about the time-averaging and potential difficulty about estimating the position of the aperture.

“The estimation of the fraction of metabolic carbon incorporated into the shell from equation 2 (reported from another previously published study – reference should be Crocker 39 and not Pape 40, by the way) relies on several poorly known parameters. One that stands out, besides d13CDIC, is the single value of -17 per mil for organic matter in nautilus. This is one measurement from one species at one point in ontogeny. I looked at the Crocker paper in depth, and I can not figure out why -17 per mil was selected. That paper presented data from 6 nautilus specimens, with different values between and within specimens. Variation within specimens can be as high as 6 per mil. There is no mention of -17 per mil in the text of the paper, so this number will have to be justified.”

RESPONSE: We have expanded the rationale for our selection of the value of �13Cmeta. We used the data from Crocker et al. to calculate average values of pre- and post-hatching siphuncular material in the six N. macromphalus reported by them. Although there is variation among the specimens, the average siphuncular �13C, either in individual specimens or overall, is essentially the same and given in the text. We calculate an overall average of -16.6 ± 1.7‰ for the metabolic �13C. Given all the uncertainties in the data, we round this to -17 ± 2‰ to calculate Cmeta from the shell �13C data in our specimens. A range in �13Cmeta of -15 to -19‰, based on the calculated standard deviation, is also plotted in Fig. 10. The value of �13Cmeta calculated using the Crocker et al. N. macromphalus data is comparable to that measured in N. pompilius 32 years later by Pape (-17.4‰), but admittedly the latter is only a single measurement. 

“The main finding is that nautilus hatch in shallower, warmer waters and move to deeper depths post-hatching is supported by the use of d18Oshell as a proxy for depth. Hope is also expressed that even though fossil nautilids can’t often be studied geochemically to reconstruct their reproductive habits/habitats, morphology could serve as a proxy for geochemical changes related to life habits. Other statements about this, however, are inconsistent. In one place, the authors state that changes in geochemistry concide with morphological parameters. Elsewhere, they state that geochemical and morphological changes are “within the range of fluctuation and thus difficult to correlate.” Later, they state that “detectable morphological changes are not clearly reflected” in the geochemistry. Which is it? This message is very muddled, and the significance for future work is not clear.”

RESPONSE: We meant that the morphological and geochemical changes coincide ONLY at the point of hatching and maturity and that the correlation is not clear in the rest of ontogeny. We clarified the text to emphasize this point.

“The aims of the paper are stated very clearly.There are a few parts that could be written more clearly. For example, what is “presence of septal approximation”? Define specialized terms for non-specialists.”

RESPONSE: We added an explanation for the somewhat technical terms.

“Equation 2 is missing some parentheses.”

RESPONSE: The equation has been modified but was correct as written.

“There’s a typo in Fig. 6E. One of the septum numbers is missing.”

RESPONSE: We corrected the typo.

“There may be issues with the references. The authors refer to Pape 40, and there is a Pape 40, but they clearly meant Crocker 39. A standard reference check would not catch that. References should be rechecked carefully.

Please show the abbreviated equations in the Fig. 6 caption in the parentheses instead of the acronyms. There’s no point in showing the acronym for the term since the term is already spelled out in the same sentence. However, what a reader may have trouble recalling are the equations for each index. It’s very frustrating as a reviewer or reader to have do a scavenger hunt to be confused about what the terms mean and then have to search the text. In this case, I had to open three windows side by side to figure it out, one for this figure, one for the text definitions of indices, and a third for the figure showing the measurements used in the indices. Ideally, any reader should be able to look at a graph and caption and get it.”

RESPONSE: As noted above, both Crocker et al. and Pape are relevant and are referenced in the discussion of the estimation of �13Cmeta. We retain both references. We removed the acronym in the caption for Fig. 6 and added abbreviated equations.

“Also in Fig. 6, should septal position index be siphuncle position index? The former is not defined in the methods, but the latter is.”

RESPONSE: We corrected the labels in Fig. 6. 

“Also in Fig 6A and B, the septal spacing graph is hard to compare to the other graphs because x-axis is different. That makes it difficult to look at the graphs an intuitively know whether all the morphological changes are happening at the same point in ontogeny or not.”

RESPONSE: We corrected the inset plots so that both graphs can share the same x-axis.

---

## [Editor Report · Decision Letter 1]

27 Jun 2022

Refining the habitat of Nautilus macromphalus based on knowledge of the δ18O and δ13C of the shell and the temperature and δ18O of the water column in New Caledonia

PONE-D-21-35042R1

Dear Dr. Tajika,

We’re pleased to inform you that your manuscript has been judged scientifically suitable for publication and will be formally accepted for publication once it meets all outstanding technical requirements.

Kind regards,

Geerat J. Vermeij

Academic Editor

PLOS ONE

Additional Editor Comments (optional):

The paper is now accepted except that I insist on a shortening of the title. I recommend: Isotopic evidence concerning the habitat of Nautilus macromphalus in New Calednonia.
---

## [Editor Report · Acceptance letter]

1 Jul 2022

PONE-D-21-35042R1 

Isotopic evidence concerning the habitat of Nautilus *macromphalus* in New Caledonia 

Dear Dr. Tajika:

I'm pleased to inform you that your manuscript has been deemed suitable for publication in PLOS ONE. Congratulations! Your manuscript is now with our production department. 

Kind regards, 

on behalf of

Dr. Geerat J. Vermeij 

Academic Editor

PLOS ONE